# Genetic separation of Brca1 functions reveal mutation-dependent Polθ vulnerabilities

John J. Krais [1,8] ✉, David J. Glass[1,2], Ilse Chudoba[3], Yifan Wang[1], Wanjuan Feng[4], Dennis Simpson[4], Pooja Patel[1], Zemin Liu[5], Ryan Neumann-Domer[5], Robert G. Betsch[1], Andrea J. Bernhardy[1], Alice M. Bradbury[1], Jason Conger[1], Wei-Ting Yueh[1], Joseph Nacson[1], Richard T. Pomerantz [6], Gaorav P. Gupta [7], Joseph R. Testa [5,7] & Neil Johnson [1] ✉

Homologous recombination (HR)-deficiency induces a dependency on DNA polymerase theta (Polθ/*Polq*)-mediated end joining, and Polθ inhibitors (Polθi) are in development for cancer therapy. *BRCA1* and *BRCA2* deficient cells are thought to be synthetic lethal with Polθ, but whether distinct HR gene mutations give rise to equivalent Polθ-dependence, and the events that drive lethality, are unclear. In this study, we utilized mouse models with separate Brca1 functional defects to mechanistically define Brca1-Polθ synthetic lethality. Surprisingly, homozygous *Brca1* mutant, *Polq*⁻/⁻ cells were viable, but grew slowly and had chromosomal instability. *Brca1* mutant cells proficient in DNA end resection were significantly more dependent on Polθ for viability; here, treatment with Polθi elevated RPA foci, which persisted through mitosis. In an isogenic system, *BRCA1* null cells were defective, but *PALB2* and *BRCA2* mutant cells exhibited active resection, and consequently stronger sensitivity to Polθi. Thus, DNA end resection is a critical determinant of Polθi sensitivity in HR-deficient cells, and should be considered when selecting patients for clinical studies.

BRCA1 functions in homologous recombination (HR) by counteracting the 53BP1-Shieldin protein complex, thereby promoting DNA end resection[1,2]. Of equal significance, BRCA1-BARD1 recruits PALB2-BRCA2-RAD51 to double-stranded DNA breaks (DSBs) to facilitate efficient RAD51 filament formation[3–6]. *Brca1* loss-of-function mutations that severely impair HR often induce embryonic lethality in mice. BRCA1 is known to compete with 53BP1-Shieldin complex for post-translationally modified histones[7,8]. Remarkably, intercrossing *53bp1* knockout (KO) alleles rescues *Brca1* mutant mouse development[9]. Rescue occurs via restored HR DNA repair[1]; however, whether additional DSB repair pathways contribute to this rescue is unknown. 53bp1

loss-of-function enhances resection, which can direct repair pathway choice toward end-joining pathways that require single-stranded (ss) DNA overhangs[10,11].

Polθ is a DNA repair enzyme that facilitates synapsis and the subsequent extension of 3′-ssDNA overhangs generated by resection of a DSB site by utilizing microhomology tracts[12–14]. This DNA repair mechanism can lead to templated nucleotide insertions or deletions flanked by short regions of sequence identity, i.e., microhomology, and is referred to as microhomology-mediated end joining (MMEJ), alterative end joining (Alt-EJ), or theta-mediated end joining (TMEJ)[15]. Polθ activity is thought to protect against extensive deletions, and limit

[1]Nuclear Dynamics Program, Fox Chase Cancer Center, Philadelphia, PA 19111, USA. [2]Temple University, Lewis Katz School of Medicine, Philadelphia, PA 19140, USA. [3]MetaSystems Probes, GmbH, Industriestr, 68804 Altlussheim, Germany. [4]Department of Radiation Oncology, Lineberger Comprehensive Cancer Center, University of North Carolina, Chapel Hill, NC 27599, USA. [5]Cytogenetics Laboratory, Fox Chase Cancer Center, Philadelphia, PA 19111, USA. [6]Thomas Jefferson University, Sidney Kimmel Cancer Center, Department of Biochemistry and Molecular Biology, Philadelphia, PA 19107, USA. [7]Cancer Control and Prevention Program, Fox Chase Cancer Center, Philadelphia, PA 19111, USA. [8]Present address: Division of Oncology, Department of Medicine, Washington University School of Medicine, St. Louis, MO 63110, USA. ✉e-mail: krais@wustl.edu; neil.johnson@fccc.edu

loss of heterozygosity, at the cost of deletion or addition of a few nucleotides at repair sites[16,17]. TMEJ signatures at DSB sites can be distinguished by the presence of deletions formed through the annealing of 2–6 bp flanking microhomology, and/or templated insertions at DSB sites. Such sequences are associated with Signature 3, which is frequently observed in HR-defective cancers[12,18–20]. Indeed, seminal studies revealed a synthetic lethal relationship between Polθ and HR repair encoding genes[21–24].

*BRCA1* mutation-containing cancers demonstrate varying responses to PARP inhibitor (PARPi) therapy, in part due to differences in the severity of HR malfunction[25]. The presence of BRCA1 hypomorphic proteins lacking various functional domains[26–29], as well as mutations in DNA end resection genes such as *S3BP1,* can promote HR and PARPi resistance[30,31]. In contrast to PARPi, mutations in *53BP1* and other DNA end resection regulatory proteins increase cellular sensitivity to Polθ inhibitors (Polθi)[11,32,33]. However, the impact of distinct *BRCA1* mutations and functional defects on cellular and organismal Polθ dependency is unknown. Moreover, several *BRCA1* and *BRCA2* mutant cancer cell lines show limited sensitivity to pharmacologic inhibition of Polθ activity[32,34]. Biomarkers that predict Polθi sensitivity within *BRCA1/2* mutation carriers would be important for identifying optimal patient populations and informing clinical trials.

Here, we set out to assess Polθ dependency in mice and derived cell lines with mutations that disrupt distinct Brca1 functions, specifically in DNA end resection versus Palb2-Brca2-Rad51 recruitment. Furthermore, we characterized the effects of mutations in conjunction with loss of 53bp1, revealing new insights into cellular and organismal Polθ dependency, with relevance for the clinical application of Polθi.

## Results

### Genetic context governs Polθ-dependent viability

To test if distinct Brca1 functional defects confer similar Polθ dependencies, we used previously established murine *Brca1^{ΔII}* and *Brca1^{CC}* mutation-containing alleles. The *Brca1^{ΔII}* allele produces a truncated protein that is defective for counteracting 53bp1, resulting in a failure to initiate DNA end resection[1]. In contrast, the *Brca1^{CC}* protein contains a coiled-coil (CC) domain deletion[35], which has little impact on DNA end resection but is highly disruptive for Palb2 binding, resulting in failed Palb2-Brca2-Rad51 loading at DSBs (Fig. 1A). Homozygous *Brca1^{ΔII/ΔII}* and *Brca1^{CC/CC}* MEFs with *53bp1^{+/+}* or *53bp1^{-/-}* genotypes were readily ascertained from crosses of heterozygous transgenic mice, and were subsequently transformed with SV40. Each genotype demonstrated the expected Brca1 and 53bp1 protein expression patterns (Fig. 1B).

To determine whether Polθ is essential for viability, we used a CRISPR/Cas9 strategy targeting *Polq* exon 4 followed by puromycin selection and sequencing of single-cell colonies to identify *Polq^{-/-}* cells (Fig. 1C). Clones were either wild-type or contained in-frame or frameshifting mutations in one or both alleles. We discounted clones that had no mutation due to a possible lack of editing activity. The number of clones that had frameshift mutations in both alleles, generating *Polq^{-/-}* genotypes, relative to the total number of clones with any *Polq* mutation, such as inframe indels or frameshift in just one allele, which may continue to support TMEJ, were quantified (Fig. 1D and Supplementary Data 1). In line with the known synthetic lethal relationship between *Brca1* and *Polq, Brca1^{+/+}, Polq^{-/-}* cells were readily detected, but only one *Brca1^{CC/CC}, Polq^{-/-}* clone was obtained. Surprisingly, the proportion of *Brca1^{ΔII/ΔII}, Polq^{-/-}* clones obtained was equivalent to that of *Brca1^{+/+}, Polq^{-/-}* cells. In *53bp1^{-/-}* cells, fewer *Brca1^{+/+},53bp1^{-/-}, Polq^{-/-}* colonies were isolated, and no *Brca1^{ΔII/ΔII}, 53bp1^{-/-}, Polq^{-/-}* or *Brca1^{CC/CC}, 53bp1^{-/-}, Polq^{-/-}* clones were obtained (Fig. 1D).

We next aimed to determine whether cellular phenotypes could be recapitulated in whole organisms. *Brca1* mutations induce embryonic lethality, therefore it is difficult to discern the impact of Polθ. However, in the setting of 53bp1 deficiency, *Brca1^{ΔII/ΔII}* mice are

born at Mendelian frequencies and develop normally, with HR restored[1,9]. In contrast, *Brca1^{CC/CC},53bp1^{-/-}* pups were born at around 50% of the expected rate and died of lymphoma with short lifespans (Supplementary Fig. 1a–d), suggesting partial rescue of HR. To investigate if Polθ contributes to 53bp1 loss-associated rescue of development, we intercrossed *Brca1,53bp1* and *Polq^{-/-}* mice. Here, *Brca1^{+/+}, 53bp1^{-/-}, Polq^{-/-}* mice were born at Mendelian ratios and developed normally (Fig. 1E and Supplementary Fig. 1e). In contrast, no live *Brca1^{CC/CC}, 53bp1^{-/-}, Polq^{-/-}* nor *Brca1^{ΔII/ΔII}, 53bp1^{-/-}, Polq^{-/-}* pups were observed (Fig. 1F), indicating that loss of Polθ activity reversed 53bp1 KO-mediated rescue of *Brca1^{CC/CC}* and *Brca1^{ΔII/ΔII}* embryogenesis. Taken together, the type of *Brca1* mutation strongly influenced *Polq*-synthetic lethality in *53bp1* wild-type cell lines, whereas when combined with *53bp1* deficiency, both *Brca1* mutations required Polθ for cellular and embryonic viability.

### Brca1 mutant, Polq^{-/-} cells have chromosomal instability

Despite retaining viability in the absence of Polθ, we hypothesized that *Brca1* mutant, *Polq^{-/-}* cells would demonstrate genome instability. The consequences of *Polq* KO were compared between *Brca1* genotypes (Supplementary Fig. 2a), focusing on *53bp1^{+/+}* cells, given we were unable to acquire *53bp1^{-/-}* derivatives. We first characterized the effects of Polθ deficiency on cell growth using clones derived in Fig. 1D. Here, *Brca1^{+/+}* and *Brca1^{CC/CC}* cells with *Polq^{-/-}* genotypes were unaffected or severely slowed, respectively. *Brca1^{ΔII/ΔII}, Polq^{-/-}* cells also grew at a modestly reduced rate compared to *Brca1^{ΔII/ΔII}, Polq^{+/+}* cells (Fig. 2A and Supplementary Fig. 2b). Of significance, *Polq^{-/-}* clones did not show PARPi resistance (Supplementary Fig. 2c), indicating that HR was not restored in *Brca1* mutant clones that can tolerate Polθ deficiency.

In line with the expected effects of SV40-induced transformation, aneuploidy was observed in karyotypes of all MEF cell lines, but limited structural abnormalities were found (Fig. 2B). Moreover, *Brca1^{+/+}, Polq^{+/+}* and *Brca1^{+/+}, Polq^{-/-}* cells had relatively few marker chromosomes, i.e., structurally abnormal chromosomes that cannot be unambiguously identified cytogenetically (Fig. 2B). In contrast, *Brca1^{CC/CC}, Polq^{-/-}* and *Brca1^{ΔII/ΔII}, Polq^{-/-}* cells had severely abnormal karyotypes, with many marker chromosomes and rearrangements (Fig. 2B), including translocation derivative chromosomes, dicentrics, deletions, and double minutes, as well as evidence of chromosomal damage, such as chromatid breaks, isochromatid breaks, and fragments (Fig. 2C and Supplementary Table 1). Interestingly, metaphases from *Brca1^{CC/CC}, Polq^{-/-}* cells routinely demonstrated greater numbers of abnormal chromosomes relative to *Brca1^{ΔII/ΔII}, Polq^{-/-}* cells (Fig. 2B and Supplementary Table 1).

To gain insight into the makeup of marker chromosomes, we performed mFISH analyses on *Brca1^{CC/CC}, Polq^{+/+}* and *Brca1^{CC/CC}, Polq^{-/-}* cells. Consistent with the G-banding studies, *Polq^{-/-}* karyotypes showed many more derivative chromosomes (structurally rearranged chromosomes generated by two or more chromosomes, e.g., the unbalanced product of a translocation) than *Polq^{+/+}* karyotypes (Fig. 2D), with most of these rearranged chromosomes appearing to be clonal (Supplementary Fig. 2d, Supplementary Data 2, 3). Strikingly, several *Brca1^{CC/CC}, Polq^{-/-}* marker chromosomes consisted of segments derived from up to 3 or even 5 distinct chromosomes (Fig. 2D, Supplementary Data 2-3). Therefore, Polθ loss exacerbates chromosomal breakage and rejoining in *Brca1* mutant cells, the consequences of which, likely manifest in slowed cell division.

### Brca1 mutations impact Polθi sensitivity

Small molecule Polθi is under development for the treatment of *BRCA1/2* mutant cancers, and genotype-conferred sensitivity profiles could be useful for identifying optimal patient groups. To assess responses to pharmacological targeting of Polθ, colony formation assays were carried out with the small molecule Polθi ART558[32]. Cell line sensitivity profiles were compared to the PARPi rucaparib.

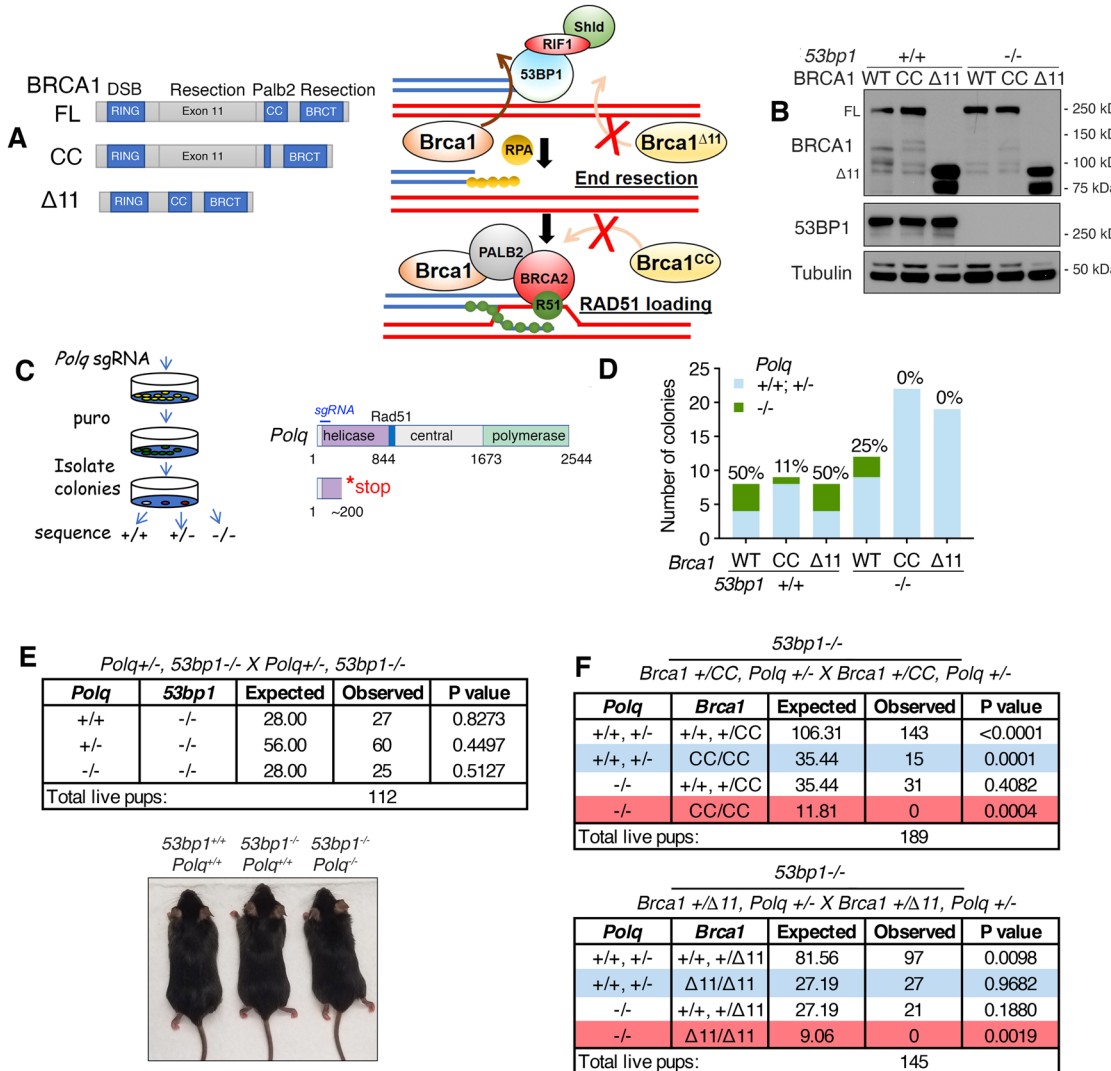

**Fig. 1 | Assessment of cellular and organismal *Polq*⁻/⁻ phenotypes. A** Cartoon showing (*left*), *Brca1* alleles and protein products with domain functions indicated; (*Right*), Brca1 functions in HR (simplified), the Brca1-Δ11 protein is defective for DNA end resection; the Brca1-CC protein is defective for Palb2-Brca2-Rad51 loading. **B** Western blotting for the indicated proteins in MEF cell lines. **C** Cartoon showing (*left*), process for generating CRISPR/Cas9 manipulated MEFs. (*Right*), Polθ protein domains and predicted effect of stop codon from sgRNA targeted exon 4 of *Polq*. **D** *Polq* gene mutations were binned according to whether frameshift mutations in both alleles (−/−) (green), or retaining at least one allele with a missense or one frameshift mutation (+/+, +/−) (blue). See Supplementary Data 1. **E** Mice with the indicated genotypes were intercrossed, and expected and observed genotypes were shown. A representative photograph of mice with the indicated genotypes is shown. See Supplementary Fig. 1e. *P* values are obtained from two-sided chi-square goodness of fit tests for the binomial of each genotype. **F** Mice with the indicated genotypes were intercrossed, and expected and observed genotypes were shown. *P* values were obtained from chi-square goodness of fit tests comparing expected and observed genotypes. Source data are provided as a Source Data file.

In *53bp1*⁺/⁺ cells, both *Brca1*^Δ11/Δ11 and *Brca1*^CC/CC were exquisitely sensitive to PARPi. In *53bp1*⁻/⁻ genotypes, *Brca1*^Δ11/Δ11 and *Brca1*^CC/CC cells showed robust and mild PARPi resistance, respectively (Fig. 3A). Of note, both *Brca1*^CC/CC, *53bp1*⁻/⁻ and *Brca1*^Δ11/Δ11 *53bp1*⁻/⁻ MEFs were more PARPi sensitive relative to wild-type MEFs. *Brca1*^Δ11/Δ11 and *Brca1*^CC/CC cells showed moderate and greater sensitivity to ART558 treatment compared to *Brca1*⁺/⁺ cells, respectively. All *53bp1*⁻/⁻ cell lines were more responsive to ART558 than their *53bp1*⁺/⁺ counterparts, with the strongest response in *Brca1*^CC/CC, *53bp1*⁻/⁻ (Fig. 3B). Thus, small molecule inhibition of Polθ using ART558 produced sensitivity profiles that largely reproduced genetic dependencies.

### *Brca1* mutations have distinct DSB outcomes
The severity of phenotypes observed with both genetic *Polq* KO and ART558 sensitivity was variable, with *Brca1*^Δ11/Δ11 cells showing mild, and *Brca1*^CC/CC cells a more severe growth slowing (Fig. 2A), or reduction in

colony formation (Fig. 3B), respectively. However, when combined with 53bp1 deficiency, both *Brca1*^Δ11/Δ11 and *Brca1*^CC/CC cell lines demonstrated genetic synthetic lethality with *Polq* KO and increased sensitivity to ART558, despite also showing increased resistance to PARPi. We next aimed to gain insight into the molecular drivers of genotype-conferred Polθ dependency. Of note, we did not observe significant differences in *Polq* mRNA expression between genotypes (Supplementary Fig. 3a).

HR and TMEJ are competing DSB repair pathways that impact sensitivity to ART558[32]. We used targeted DSB reporter assays to measure the relative activity of these pathways in our cell line panel. DSBs were induced in the *Rosa26* locus with CRISPR/Cas9 and digital droplet (dd)PCR was used to quantify pathway-specific repair products[11,36]. The inclusion of an HR donor template showed that *Brca1*^CC/CC and *Brca1*^Δ11/Δ11 cells had reduced HR activity relative to *Brca1*⁺/⁺ cells. HR repair levels significantly increased in *Brca1*^Δ11/Δ11,

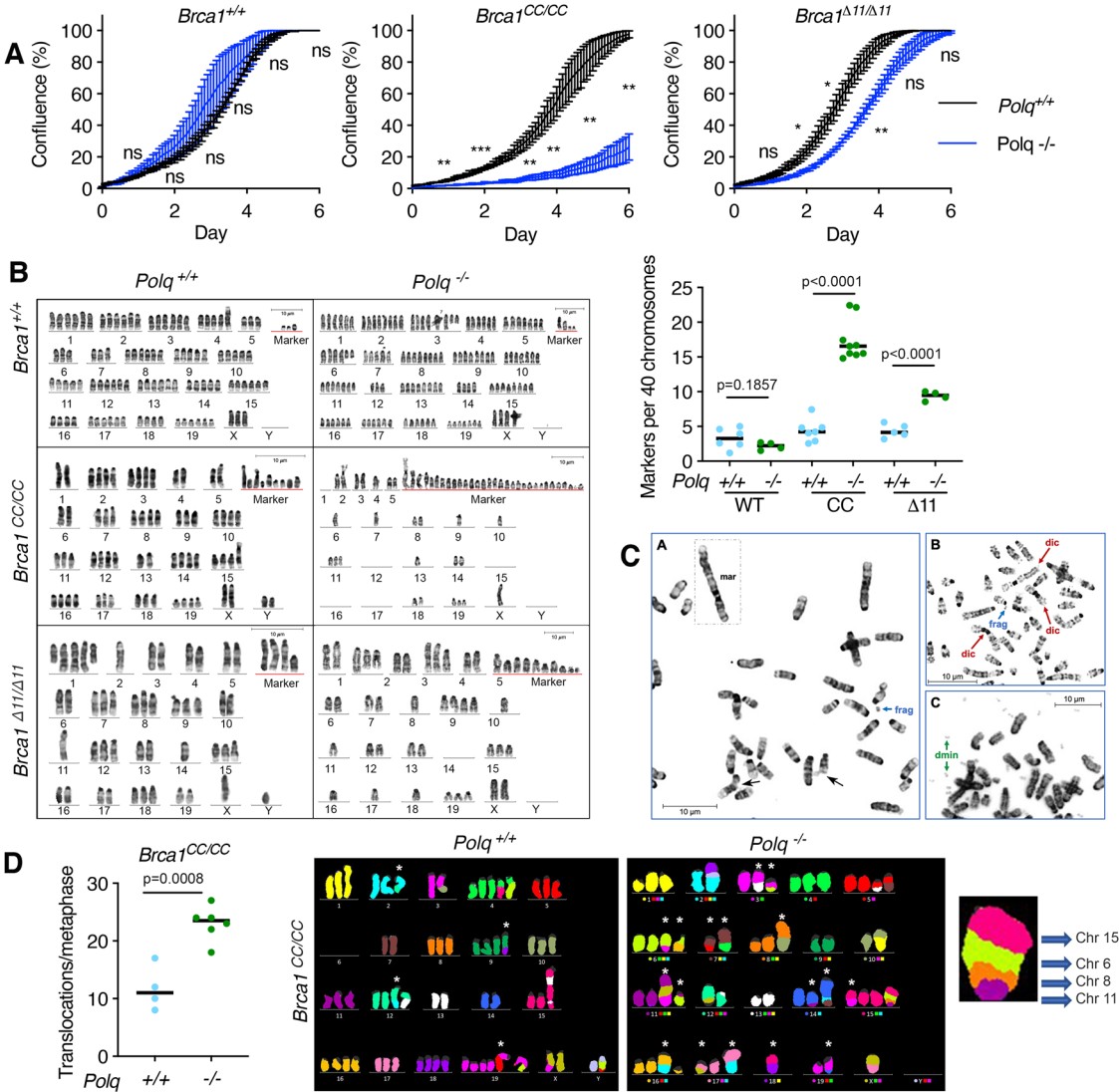

**Fig. 2 | Cell growth and chromosome analyses of *Polq*⁻/⁻ genotypes. A** Growth curves were obtained for *Polq*⁻/⁻ clones and control cells (sgGFP) for *Brca1*⁺/⁺, *Brca1*^CC/CC^, and *Brca1*^Δ11/Δ11^ MEFs based on confluence measurements obtained using a Sartorius Incucyte imager. Confluence and S.E.M. of three independent experiments are shown. ***p < 0.001; **p < 0.01; *p < 0.05; ns, not significant (unpaired, two-tailed *t* tests comparing *Polq*⁺/⁺ and *Polq*⁻/⁻ confluence on each day).
**B** Representative karyograms are shown for the indicated genotypes and marker chromosome counts are shown normalized to 40 chromosomes per metaphase. Each data point is an individual metaphase, *n* = 6, 4, 7, 9, 5, 4 (unpaired, two-tailed *t* tests). See Supplementary Table 1. **C** Example G-banded partial metaphase spreads

from *Brca1*^CC/CC^, *Polq*⁻/⁻ cells highlighting marker (mar) chromosome, fragments (frag), dicentric (dic), double minutes (dmin), and chromatid breaks (arrows) are shown from 1 experiment. **D** *Brca1*^CC/CC^, *Polq*⁺/⁺ and *Brca1*^CC/CC^, *Polq*⁻/⁻ metaphase spreads were stained using mFISH and karyograms generated. The number of translocations per metaphase were quantified, *n* = 4, 6. Representative mFISH karyograms are shown, and a magnified image of a derivative chromosome consisting of segments from four different chromosomes is highlighted, far right. * indicate clonal marker chromosomes. See Supplementary Data 2, 3 and Supplementary Fig. 2d. Source data are provided as a Source Data file.

*53bp1*⁻/⁻ MEFs, but there was only a minor increase in *Brca1*^CC/CC^, *53bp1*⁻/⁻ compared to *53bp1*⁺/⁺ counterparts (Fig. 4A). TMEJ repair products were slightly elevated in *Brca1*^CC/CC^ MEFs but reduced in *Brca1*^Δ11/Δ11^ cells, relative to *Brca1*⁺/⁺ MEFs. With *53bp1* deficiency, TMEJ activity increased across genotypes, and *Brca1*^CC/CC^, *53bp1*⁻/⁻ MEFs had markedly higher TMEJ levels relative to *Brca1*^Δ11/Δ11^, *53bp1*⁻/⁻ MEFs (Fig. 4B).

We next explored DNA repair events typically associated with HR and TMEJ. DNA end resection stimulates both HR and TMEJ[11], and we confirmed previous results showing that *Brca1*^CC/CC^ are proficient, and *Brca1*^Δ11/Δ11^ cells defective for resection measured by END-seq[35], which directly measures ssDNA lengths[37]. As expected, *53bp1* KO increased resection lengths across all genotypes (Fig. 4C). RPA32 γ-irradiation-induced foci (IRIF) was used as another indirect measurement of resection and generally reflected END-seq trends (Fig. 4D). Together, these results re-enforce that RPA32 foci are reflective of DNA end

resection products, i.e., ssDNA overhangs. We confirmed low levels of Rad51 IRIF in *Brca1* mutant, *53bp1*⁺/⁺ cells. Strikingly, Rad51 IRIF was rescued in *Brca1*^Δ11/Δ11^, *53bp1*⁻/⁻ cells, but foci remained relatively low in *Brca1*^CC/CC^, *53bp1*⁻/⁻ cells (Fig. 4E). The latter observations are in line with previous work showing that in the absence of 53bp1, Brca1 hypomorphic proteins that interact with Palb2 are required for efficient Rad51 loading and HR[35,36].

We hypothesized that the ability of Brca1-Δ11 to bind Palb2[26,36], and promote Rad51 loading and HR contributed to the differential ART558 sensitivity observed between *Brca1*^Δ11/Δ11^, *53bp1*⁻/⁻ and *Brca1*^CC/CC^, *53bp1*⁻/⁻ cells (Fig. 3B). Indeed, *Brca1* siRNA decreased Brca1-Δ11 expression and reduced Rad51 IRIF (Fig. 4F). Moreover, *Brca1* siRNA further reduced the colony formation of *Brca1*^Δ11/Δ11^, *53bp1*⁻/⁻ MEFs treated with ART558 (Fig. 4G). These results confirm that in the setting of 53bp1 deficiency, the Brca1-Δ11 protein, which retains the CC

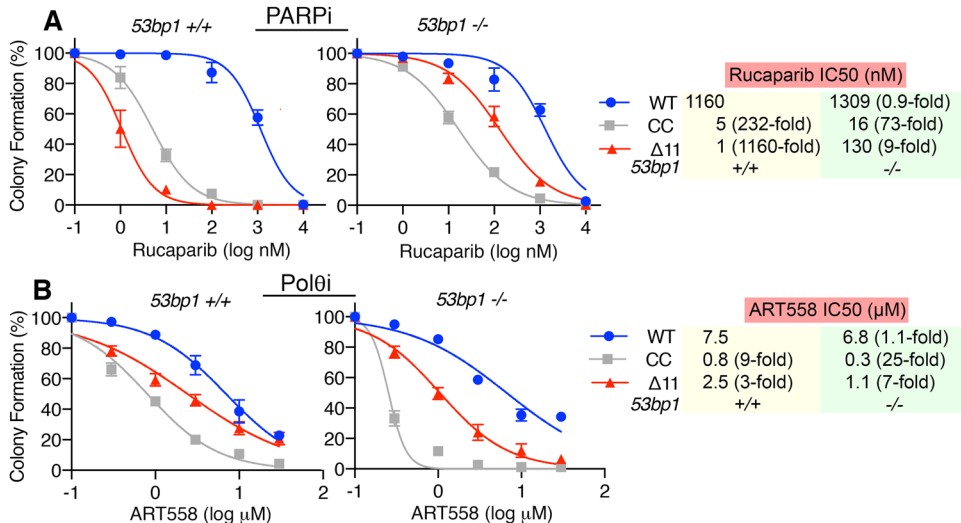

**Fig. 3 | ART558 and PARPi sensitivity profiles. A** Cells were incubated with increasing concentrations of rucaparib for 2 weeks and colony formation assessed. Mean and S.E.M. colony formation as well as mean IC50 values, $n = 4$ biological replicates. Fold changes in IC50 relative to *Brca1*[+/+], *53bp1*[+/+] (WT) are shown. **B** Cells were incubated with increasing concentrations of ART558 for 2 weeks and colony formation assessed, as for **A**. Source data are provided as a Source Data file.

domain, promotes Rad51 loading, enabling some DSBs to be repaired by HR, reducing ART558 sensitivity. Taken together, *Brca1* mutations that impact DNA end resection or the Palb2 interaction, both alone and in combination with 53bp1 deficiency, induce varying levels of HR and TMEJ activity, and consequently sensitivity to Polθi (Fig. 4H).

## RPA persists through mitosis with ART558

We next considered the consequences of DNA end resection proficiency on DNA repair outcomes during ART558 treatment. First, Western blotting measurements of γH2ax indicated that, in contrast to PARPi treatment where DNA damage was present in all cell lines, ART558 treatment only induced detectable increases in γH2ax in *Brca1*[CC/CC] cells (Fig. 5A). Unrepaired replicative DNA damage often manifests as mitotic abnormalities and we measured the consequences of ART558 treatment on mitosis using H2B-mCherry labeling, which allows visualization and the live cell imaging of condensed chromosomes. Here, ART558 treatment marginally increased the number of abnormal mitoses in *Brca1*[+/+] and *Brca1*[Δ11/Δ11] cells. In contrast, the number of abnormal mitoses and lagging chromosomes, as well as micronuclei, dramatically increased in *Brca1*[CC/CC] cells treated with ART558 (Fig. 5B and Supplementary Fig. 3b). These data suggest that ART558 minimally impacts the overall levels of spontaneous replication-associated breaks in *Brca1*[+/+] and *Brca1*[Δ11/Δ11] cells, but increases DNA damage in *Brca1*[CC/CC] cells, leading to mitotic abnormalities.

We hypothesized that our inability to detect significant levels of DNA damage with ART558 treatment in asynchronous *Brca1*[Δ11/Δ11] cells might be due to transient damage that is repaired. To test this idea, we carried out imaging of synchronized cells post-thymidine release and monitored the presence and resolution of ssDNA overhangs and DNA breaks using RPA-mCherry and GFP-MDC1, respectively. Here, *Brca1*[Δ11/Δ11] cells that were MDC1 foci-positive post-thymidine release showed a trend for resolution during mitosis. In contrast, the relative decrease in MDC1 foci was lessened in ART558 treated *Brca1*[Δ11/Δ11] cells, with daughter cells demonstrating 1.9-fold more MDC1 foci per nucleus relative to vehicle-treated cells (Fig. 5C). RPA-mCherry foci were rarely detected by live cell imaging methods in *Brca1*[Δ11/Δ11] cells.

In *Brca1*[CC/CC] cells, while most cells with MDC1 foci showed resolution during mitosis in vehicle-treated cells, ART558 treatment resulted in the persistence of foci through mitosis, with daughter cells demonstrating 2-fold more foci per nucleus (Fig. 5C). Furthermore, in

vehicle-treated *Brca1*[CC/CC] cells, most RPA foci resolved during mitosis. However, with ART558 treatment, the number of RPA foci were significantly elevated pre-mitosis and showed only partial resolution in mitosis, with daughter cells demonstrating 4.2-fold more foci per nucleus (Fig. 5C). We surmise that during ART558 treatment, *Brca1*[Δ11/Δ11] cells have limited resection of breaks that arise during replication, providing compatible NHEJ substrates, which are repaired prior to, or following mitosis. In contrast, *Brca1*[CC/CC] cells frequently resect DSBs, generating suitable TMEJ, but poor NHEJ substrates[11], and as a consequence, RPA-coated ssDNA overhangs and DNA damage persist with ART558 treatment.

DNA breaks that endure throughout the cell cycle can be aberrantly joined, potentially accounting for the increase in complex marker chromosomes and other clonal rearrangements observed in *Brca1* mutant, *Polq*[−/−] cells (Fig. 2B). We sought to determine whether loss of Polθ activity directly increases the number of chromosome rearrangements in *Brca1*[CC/CC] MEFs. To test this, we artificially induced DSBs in the *Rosa26* and *H3f3b* mouse loci using CRISPR/Cas9 in *Brca1*[CC/CC] cells that were treated with either DMSO or ART558[38]. Translocation frequency did not increase with ART558 (Fig. 5D, Supplementary Fig. 3c) and as expected, ART558 resulted in fewer junctions containing microhomology (Fig. 5E, Supplementary Fig. 3d). Interestingly, vehicle-treated cells had a median deletion size of 18.5 bp, but the median deletion in ART558 treated *Brca1*[CC/CC] cells was 86 bp (Fig. 5E). Additionally, insertions were observed with ART558 treatment (Supplementary Fig. 3d). We speculate larger deletions induced by ART558 treatment could be due to resection associated single-strand annealing (SSA) repair, or cleavage of overhangs to generate NHEJ substrates. Thus, when DNA end resection is active, loss of Polθ activity results in persisting DNA breaks, with some chromosome fragments eventually aberrantly joined, perhaps by SSA or NHEJ.

## HR gene mutations confer distinct Polθi sensitivity

During HR, BRCA1 functions to promote DNA end resection and RAD51 loading. In contrast, PALB2 and BRCA2 are dispensable for resection, but required for RAD51 loading and strand invasion[39]. Because resection-proficient mouse cells showed sustained DNA damage in the presence of ART558, we hypothesized that different HR gene mutations might have distinct ART558 sensitivity profiles. To test this idea, we compared the effects of BRCA1 versus PALB2 deficiency in an isogenic human system[40]. MDA-MB-436 TNBC cells harbor a hemizygous

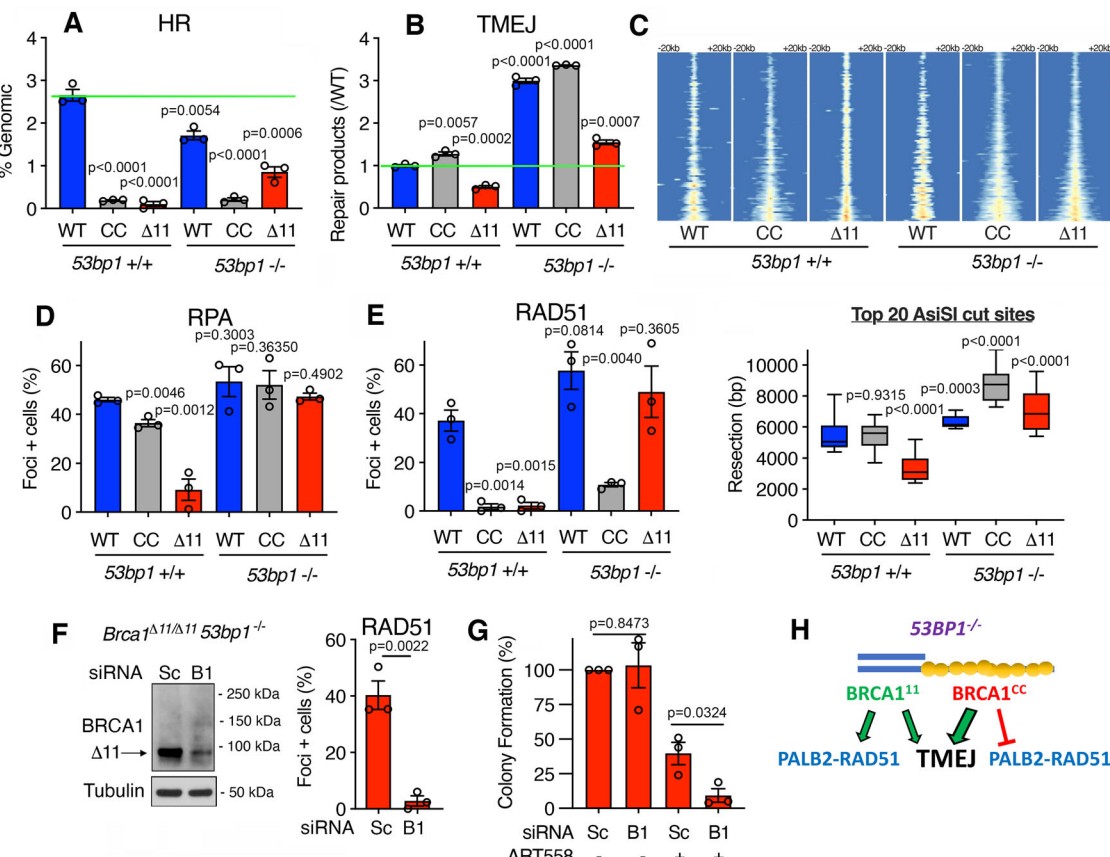

**Fig. 4 | Genotype-conferred DNA repair profiles. A** HR repair was measured by ddPCR following Cas9-induced DSB generation in cells of the indicated genotypes. HR measurements are quantifications of repair products generated from a homologous template as a proportion of the non-templated sequence and presented as a percentage of genomic DNA copies determined by a control reaction for $n = 3$ biological replicates. Mean and S.E.M. are shown. **B** TMEJ was measured by ddPCR for 3 distinct repair products, summed, and normalized to the TMEJ products detected in wild-type MEFs for $n = 3$ biological replicates. Mean and S.E.M. are shown. **C** END-seq was used to quantify resection for MEFs expressing the nuclease AsiSI with the indicated genotypes. (Top) Heatmaps of normalized, binned END-seq signal centered at AsiSI sites and extending 20 kb in both directions. (Bottom) Resection distances were calculated for each AsiSI site and quantifications shown by Tukey box and whisker plot for the top 20 resected sites for each sample, with the box drawn from the 1st to 3rd quartile and median indicated with a line. **D** Cells were (2 Gy) γ-irradiated and after 6 h pre-extracted, fixed, and assessed for Rpa foci formation using immunofluorescence staining. Mean and S.E.M. percentage of cells with more than 5 foci for $n = 3$ biological replicates. **E** Cells were assessed for Rad51 foci as described in **D**. **F** $Brca1^{\Delta11/\Delta11}$, $S3bp1^{-/-}$ cells were treated with siRNA targeting the Δ11 isoform of Brca1 (B1) or scrambled control siRNA (Sc). Left, western blotting confirmed knockdown of the Brca1-Δ11 isoform. Right, Cells were assessed for Rad51 foci formation as in **D**. Mean and S.E.M. percentage of cells with more than five foci for $n = 3$ biological replicates. **G** Cells from **F** were seeded in the presence of DMSO or 1 μM ART558 and the indicated siRNA and colony formation were measured. Colony counts were normalized to DMSO-treated controls for independent experiments and mean and S.E.M were shown for $n = 3$ biological replicates. **H** Model demonstrating the interplay between $53bp1$ and $Brca1$ mutations. Statistical significance was assessed by unpaired, two-tailed $t$ tests. Source data are provided as a Source Data file.

5396 + 1 G > A mutation, and we previously confirmed no BRCA1 protein is detectable by western blot[30,40]. Thus, MDA-MB-436 ($BRCA1^{-/-}$, $PALB2^{+/+}$) and a complemented add back version ($BRCA1^{-/-}$, $PALB2^{+/+} + BRCA1$) were used to generate stable $PALB2$ KO clones by CRISPR/Cas9, without ($BRCA1^{-/-}$, $PALB2^{-/-}$) or with BRCA1 addback ($BRCA1^{-/-}$, $PALB2^{-/-} + BRCA1$) (Fig. 6A).

As expected, in the absence of BRCA1, DNA end resection measured by RPA32 IRIF was diminished, but BRCA1 add back restored resection, regardless of $PALB2$ status (Fig. 6B). In line with the known functions of PALB2, RAD51 IRIF was restored in $BRCA1^{-/-}$, $PALB2^{+/+} + BRCA1$ cells, but in the absence of PALB2, the re-expression of BRCA1 was insufficient to restore RAD51 foci in $BRCA1^{-/-}$, $PALB2^{-/-} + BRCA1$ cells (Fig. 6C). PARPi sensitivity also reflected RAD51 loading capabilities between cell lines (Fig. 6D), confirming the known functions of BRCA1 and PALB2 within HR. $BRCA1^{-/-}$, $PALB2^{+/+}$ and $BRCA1^{-/-}$, $PALB2^{-/-}$ cells showed increased ART558 sensitivity relative to HR proficient $BRCA1^{-/-}$, $PALB2^{+/+} + BRCA1$ cells. However, while both $BRCA1^{-/-}$, $PALB2^{-/-}$ and $BRCA1^{-/-}$, $PALB2^{-/-} + BRCA1$ cells are HR defective, end resection was restored with BRCA1 add back, and cells demonstrated greater

ART558 sensitivity (Fig. 6E and S4). Similar outcomes were observed in $BRCA1^{-/-}$, $BRCA2^{-/-}$ cells complemented with BRCA1 (Supplementary Fig. 4). Taken together, these results suggest that the underlying functional HR gene defect and the ability to perform end resection are determinants of Polθ sensitivity (Fig. 6F).

## Discussion

HR repair gene mutations often impart a dependence on alternative DSB repair pathways, providing therapeutic opportunities that selectively target cancer cells. Cancers with $BRCA1$ and $BRCA2$ mutations are thought to be highly dependent on Polθ activity for their survival[21,22]. Furthermore, the hyperactivation of DNA end resection, through the loss of proteins that inhibit this pathway, also induces Polθ-dependent survival[11,32,33]. Thus, our understanding of the interplay between HR gene mutations, loss of end resection factors, and susceptibility to Polθi is continuing to emerge. The presence of hypomorphic BRCA proteins, which lack certain domains and functions, while retaining others, adds complexity to existing paradigms. In the current work, we show that the underlying HR gene mutation, and its effect on DNA end

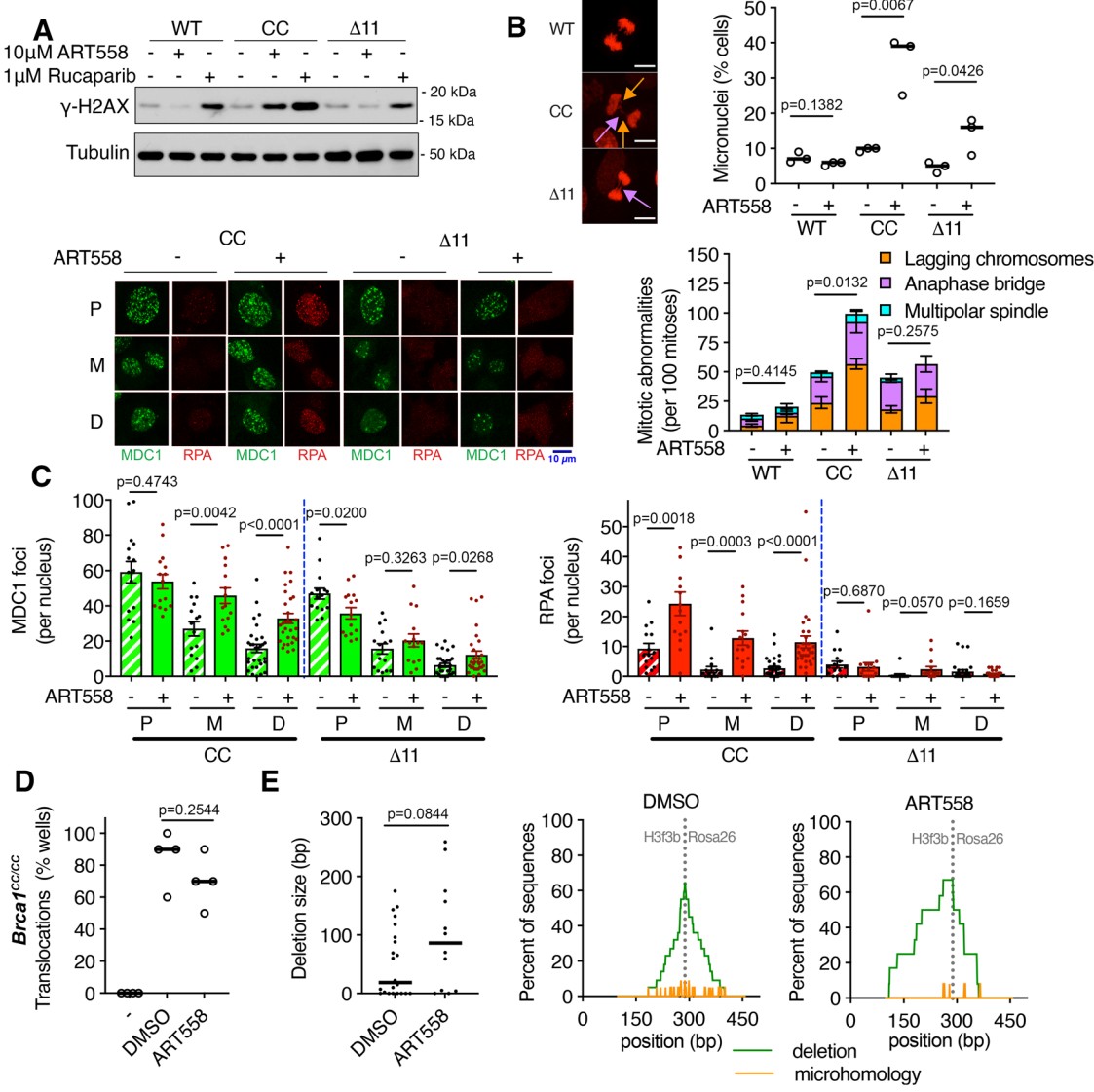

**Fig. 5 | DNA damage and translocations with ART558. A** *Brca1*[+/+], *Brca1*[CC/CC], and *Brca1*[Δ11/Δ11] MEFs were treated with DMSO, ART558, or rucaparib for 24 h, and whole cell extracts were collected to assess γH2ax by western blotting. Representative blots from three independent experiments are shown. **B** MEFs expressing H2B-mCherry were tracked after 72 h of DMSO or 10 μM ART558 treatments. (Top) The percentage of nuclei with micronuclei present are shown for individual replicates (bar is the median). (Bottom) Individual mitoses were classified as normal or abnormal with mean and S.E.M. percentages for *n* = 3 biological replicates. Representative images and examples of lagging chromosomes (orange arrow) and anaphase bridges (pink arrow) are shown. Scale bar is equal to 10 μm. See Supplementary Fig. 3b for additional images. **C** *Brca1*[CC/CC] and *Brca1*[Δ11/Δ11] MEFs were treated with thymidine for 16 h and then released into fresh media. DMSO or 10 μM ART558 was added 2 h after release and live cells were imaged for GFP-MDC1 and RPA2-mCherry. MDC1 and RPA2 foci were quantified for 15 individual cells that entered mitosis with foci present over three independent experiments. The mean and S.E.M. number of foci are shown prior to mitosis (P), during mitosis (M), and in daughter cells (D). Representative images are shown. **D** Translocation frequency in *Brca1*[CC/CC] cells was determined after inducing breaks in *Rosa26* and *H3f3b* loci. Cells growing in 96 well plates with DMSO or 10 μM ART558 were compared to wells with no break induction (−). The average percentage of wells with detectable translocation products is shown for *n* = 4 biological replicates (bar is median). See Supplementary Fig. 3c. **E** Breakpoint junctions for individual wells from **D** were sequenced and aligned to the predicted translocation product. (Left) Deletion size was determined for each sequence. (Right) The frequency of deletions (green) and microhomologies (orange) are shown at each coordinate of the predicted translocation product. If multiple sequence traces were detected and could not be deconvolved, those sequences were excluded from analyses, see Supplementary Fig. 3d. Statistical significance was assessed for the indicated comparisons by unpaired, two-tailed *t* tests. Source data are provided as a Source Data file.

resection, is a crucial determinant of cellular TMEJ addiction and the response to Polθi. Furthermore, that synthetic sickness may be a more appropriate description of the Brca1 and Polθ genetic interaction, given Polθ loss could be tolerated for viability in *Brca1* mutated, 53bp1 wild-type cells.

Surprisingly, *Brca1*[Δ11/Δ11], *Polq*[−/−] cells were readily generated, but grew more slowly. Moreover, *Brca1*[Δ11/Δ11] cells showed moderate ART558 sensitivity. The mouse *Brca1*[Δ11] allele expresses a truncated protein that lacks exon 11. *Brca1*[Δ11] cells and mice have been extensively

characterized and shown to be defective in counteracting 53bp1 and promoting DNA end resection[1]. However, because microhomology can often be found within ssDNA overhangs that are less than 20 base pairs[41], we speculate that a portion of DSBs have long enough overhangs to engage Polθ, accounting for the mild ART558 sensitivity observed in *Brca1*[Δ11/Δ11] cells. In human cancers, *BRCA1* frameshift mutations located in exon 11 account for up to one-third of all pathogenic mutations (https://research.nhgri.nih.gov/projects/bic/index.shtml). Cancer cells with these mutations are also capable of

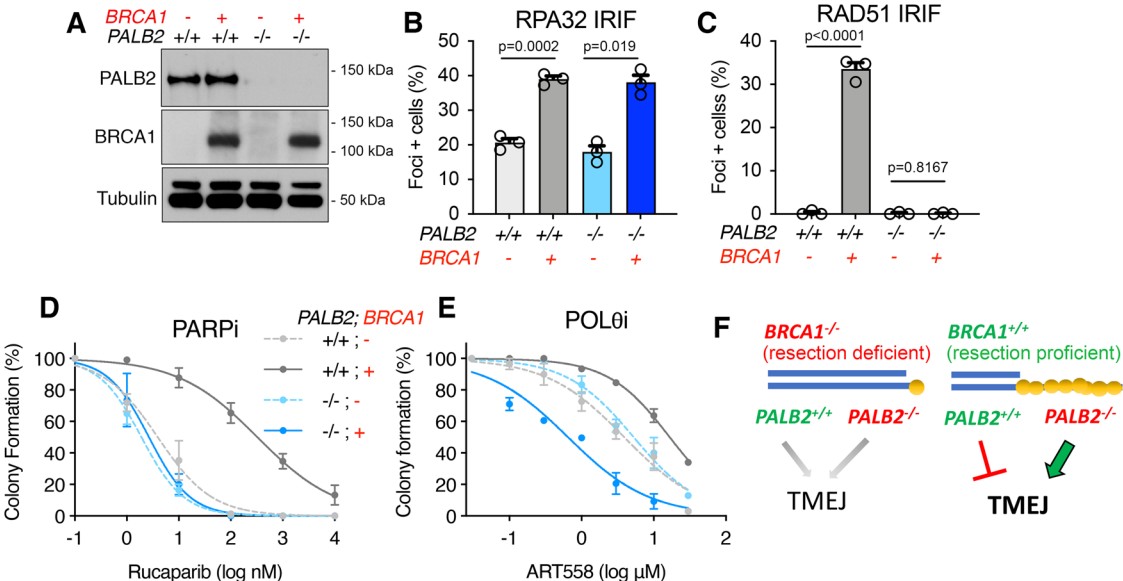

**Fig. 6 | HR deficiencies and ART558 sensitivity. A** MDA-MB-436 clones with control sgRNA targeting BFP, or *PALB2* knockout with and without ectopic BRCA1 addback. Expression of the indicated proteins was assessed by western blotting. **B** RPA foci formation was measured in geminin-positive cells 8 h after 10 Gy γ-irradiation. Cells with >10 RPA foci were counted as positive. Mean and S.E.M. shown for $n = 3$ biological replicates. **C** As for B, cells with 5 or more RAD51 foci were counted positive. **D** Cells were seeded into increasing concentrations of rucaparib and colony formation was measured after 2 weeks. Mean and S.E.M from $n = 3$ biological replicates. **E** Cells were seeded into increasing concentrations of ART558 and colony formation was measured after 2 weeks. Mean and S.E.M from $n = 3$ biological replicates. See Supplementary Fig. 4. **F** Model depicts TMEJ usage in BRCA1 proficient and deficient settings. We propose that BRCA1 deficiency leads to moderate TMEJ dependence due to minimal DNA end resection. In contrast, *BRCA1*[+/+] cells are proficient in DNA end resection resulting in hyper-dependence on TMEJ in the absence of RAD51 loading caused by PALB2 or BRCA2 mutations. Source data are provided as a Source Data file.

expressing an exon 11 deficient isoform through alternative splicing and removal of exon 11, and this protein is similarly defective for promoting resection[26]. Furthermore, cells with mutations that cause a complete loss of BRCA1 protein expression are also defective for DNA end resection[30]. Thus, our data suggest that a significant portion of patients with *BRCA1* mutation-containing cancers might show a limited response to Polθi monotherapy.

*Brca1*[CC/CC], *Polq*[−/−] cells had phenotypes that were more severe, with greater numbers of chromosome alterations, including numerous marker chromosomes. We developed the *Brca1*[CC] allele to demonstrate the importance of the Brca1-Palb2 complex for Brca2-Rad51 loading, establishing that Brca1 has additional functions in HR beyond DNA end resection that cannot be readily bypassed[35]. However, *BRCA1* CC domain missense mutations in patients with cancer are relatively uncommon. Nonetheless, the *Brca1*[CC] allele is a useful tool to separate Brca1 functions in HR, and is phenotypically similar to Palb2 and Brca2-deficient cells, with proficient DNA end resection, but severely disrupted Rad51 loading. Hence, we extended our findings to show that in human isogenic cells, PALB2- and BRCA2-deficient cells were more sensitive to ART558 than BRCA1-deficient cells. However, sensitivity was dependent on the presence of BRCA1, which was required to promote DNA end resection. The implications of these data are that cancers with *PALB2* and *BRCA2* inactivating mutations may be the optimal group for Polθi monotherapy. We acknowledge that some *BRCA1* cancers are responsive to Polθi, although loss of end resection factors could contribute[32,33,42]. Furthermore, in future studies, it will be important to test additional Polθi such as novobiocin, as well as determine possible differences between pharmacological inhibition and genetic KO of Polθ.

The biological mechanisms by which Polθ enzymatic activities support the viability of HR-deficient cells are undergoing investigation, and include roles in replication fork gap filling[34,43,44], and DSB repair[15]. In our experiments, we did not detect elevated γH2ax in asynchronous *Brca1*[Δ11/Δ11] cells with ART558 treatment. We speculate that limited resection of DSBs provides compatible substrates for NHEJ, which

repairs DNA damage in interphase, and circumvents the requirement for TMEJ- repair. TMEJ was recently shown to be active in mitosis[45,46], which is supported by our results, indicating that DSBs remain present in cells undergoing mitosis during ART558 treatment. In end resection proficient cells, we observed elevated RPA32 in pre-mitotic cells that persisted to daughter cells with ART558, suggesting that TMEJ is also active prior to mitosis, where it may be required to strip RPA from ssDNA[47].

*Polq*[−/−] cells were previously shown to have a reduced incidence of translocations between loci subject to CRISPR-based cutting[21]. In contrast, *Brca1*[CC/CC], *Polq*[−/−] cells harbored a plethora of rearranged chromosomes. We predict that in *Brca1*[CC/CC], *Polq*[−/−] cells, rearrangements accumulate due to failed repair of endogenous replication-associated DSBs, and accumulating chromosome fragments are eventually randomly joined as cells continue to proliferate. Differences could also be due to replication-associated single-ended DSBs versus CRISPR-induced two-ended DSBs. *Brca1*[CC/CC] cells treated with ART558 also had larger deletions at translocation junction sites, potentially due to active resection, SSA, or nucleases that adapt ends for NHEJ. It remains to be seen whether patient tumors will demonstrate extensive chromosome interchanges post-Polθi treatment. We speculate implications of such Polθi-induced chromosomal alterations could include cGAS-STING activation, which could prime tumors for immunotherapy[48,49].

Predicting Polθ sensitivity amongst *BRCA1* mutation carriers is complicated by other genetic factors. For example, *Brca1*[Δ11/Δ11], *53bp1*[−/−] cells were more sensitive to ART558 than *Brca1*[Δ11/Δ11], *53bp1*[+/+] cells. These results follow those in cancer cell lines and PDX models with similar genetic aberrations[32,33]. In line with this, genetic crosses of mice established that *53bp1* KO-mediated rescue of *Brca1* mutant mouse development requires both TMEJ and HR. Here, intercrossing *Polq*[−/−] mice induced embryonic lethality in both *Brca1*[Δ11/Δ11], *53bp1*[−/−] and *Brca1*[CC/CC], *53bp1*[−/−] genotypes. However, the balance between pathway usage could vary between *Brca1* mutations, as suggested by mouse phenotypes, where *Brca1*[CC/CC], *53bp1*[−/−] mice have short lifespans and develop lymphoma, whereas *Brca1*[Δ11/Δ11], *53bp1*[−/−] mice have normal life-

spans. In derived cell lines, it was apparent that $Brca1^{CC/CC}$, $53bp1^{-/-}$ cells were more sensitive to ART558 than $Brca1^{\Delta11/\Delta11}$, $53bp1^{-/-}$ cells. Here, the failure to efficiently recruit Rad51 and engage HR was responsible for Polθ hypersensitivity in $Brca1^{CC/CC}$, $53bp1^{-/-}$ cells. Thus, Polθ addiction occurs under conditions of active resection and inefficient Rad51 loading, and is exacerbated by hyper-resection.

Taken together, our study establishes cellular DNA end resection proficiency as a critical feature of the HR- Polθi synthetic lethal paradigm. Additionally, persisting ssDNA-coated RPA could be a source of Polθi-induced chromosomal rearrangements and cellular toxicity. The dissection of specific protein functions within HR revealed that Polθi monotherapy may provide greater benefit for cancers with *PALB2* and *BRCA2* mutations relative to those with *BRCA1* mutations. Our data supports the study of RPA foci as a possible predictive biomarker for Polθi sensitivity in cancers with HR gene mutations. Clinical trials underway combine Polθi with PARPi or DNA-damaging agents; our work, along with others, supports this strategy, particularly for those patients carrying *BRCA1* mutations. Moving forward, specific mutation-to-function relationships should be considered in the application of DNA repair inhibitor therapeutics.

## Methods

### Mouse breeding and genotyping
All experiments involving animals were approved by the Fox Chase Cancer Center (FCCC) Institutional Animal Care and Use Committee and housed in standard conditions. The generation of the $Brca1^{CC}$ mouse allele is previously described[35]. We obtained mice harboring *Polq* mutant, *53bp1* mutant from Jax labs, and $Brca1^{\Delta11}$ alleles from the NCI. Mice were from mixed backgrounds. Genotyping of mice was performed with DNA extracted with 50 mM NaOH. $Brca1^{CC}$ genotypes were determined by PCR amplification using the following primers: forward GAGGAGCTACAGTGAAGCAGATCT, reverse TATGA-CAAGGCACGTGTGGA. Subsequently, the 1.2 kb PCR product was subjected to an EcoNI restriction digest of DNA from wild-type mice showed 1 band, whereas DNA from heterozygous mice had 3 bands, and homozygous mutant had 2 bands. $Brca1^{\Delta11}$ mice were genotyped using two PCR reactions that indicate the presence of wild-type (465 bp band) or the Δ11 allele (-650 bp band) using the following primers: common forward CTGGGTAGTTTGTAAGCATGC, wild-type reverse CAATAAACTGCTGGTCTCAGG, Δ11 reverse CTGCGAG-CAGTCTTCAGAAAG. *53bp1* mice were genotyped using two PCR reactions that indicate the presence of wild-type (-450 bp band) or the *53bp1* allele (-450 bp band) using the following primers: wild-type forward CTCCAGAGAGAACCCAGCAG, mutant forward CTAAAGCG-CATGCTCCAGAC, common reverse GAACTTGGCTCACACCCATT. *Polq* mice were genotyped using two PCR reactions, one to detect the wild-type allele (-200 bp band) and one to detect the mutant allele (-300 bp band) with the following primers: wild-type forward TGCAGTGTACAGATGTTACTTTT, wild-type reverse TGGAGGTAG-CATTTCTTCTC, mutant forward TCACTAGGTTGGGGTTCTC, mutant reverse CATCAGAAGCTGACTCTAGAG.

### MEF generation, Polq knockout, and cell culture
MEFs were generated and transduced with SV40 large T antigen as previously described and maintained in Dulbecco's Modified Eagle's Medium with 4.5 g/L glucose, 15% fetal bovine serum, 1× GlutaMAX, 1× MEM nonessential amino acids, 1 mM sodium pyruvate, 1× penicillin/streptomycin[35]. MEFs were tested for Mycoplasma using the Lonza MycoAlert Plus kit (Lonza, LT07-710) and genotypes were confirmed using the assays described above.

CRISPR/Cas9 was used to generate *Polq* KO MEFs in $Brca1^{+/+}$, $Brca1^{CC/CC}$, and $Brca1^{\Delta11/\Delta11}$ backgrounds. The following single guide RNA sequences were cloned into the lentiCRISPRv2 backbone containing Cas9 (Addgene plasmid#52961, gift of Feng Zhang): *Polq*: GCTGGTCAATCGCCTCATTG. Lentivirus was generated using

HEK293T cells and used to transduce $Brca1^{+/+}$, $Brca1^{CC/CC}$, and $Brca1^{\Delta11/\Delta11}$ MEFs. Cells were first subject to puromycin selection then single cell clones were isolated, screened by Sanger sequencing, and validated by next-generation amplicon sequencing using the following primers: Polq forward CCTCACCAACGGGACAGTTC, Polq reverse CCTTCCCA-TATGTGCCATGCT. The alleles identified by next-generation sequencing at frequencies >1% are shown.

### Western blotting
Whole-cell extracts were generated by collecting cells in RIPA buffer with protease and phosphatase inhibitors (Millipore 524624, 524625, 539131). Nuclear extracts were obtained from cells using the NE-PER Nuclear and Cytoplasmic Extraction Kit (Thermo Fisher Scientific, 78833) following manufacturer instructions. Protease and phosphatase inhibitors were included in cytoplasmic and nuclear extraction buffers. Protein concentrations were normalized using a BCA assay according to manufacturer instructions (Thermo Fisher Scientific, 23227). Lysates were separated by SDS-PAGE and transferred to a PVDF membrane. Membrane blocking was performed for 1 h at room temperature using 5% nonfat milk in TBST, followed by overnight incubation at 4 degrees with the primary antibody in 5% nonfat milk. The following primary antibodies and dilutions were used for mouse lysates: Brca1 (R&D Systems, MAB22101, 1:1000), 53bp1 (Novus Biologicals, NB100-305SS, 1:1000), γH2ax (Millipore Sigma, 05-636, 1:1000), Tubulin (Cell Signaling, 2148, 1:2000). The following primary antibodies were used for human cancer cell line lysates: BRCA1 (Millipore Sigma, OP92, 1:500), PALB2 (Bethyl Laboratories, A301-246A, 1:2000), 53BP1 (Millipore Sigma, MAB3802, 1:1000), Tubulin (Cell Signaling, 2148, 1:2000). HRP-conjugated secondary antibodies (Cytiva NA931V, NA934V) were incubated with membranes for 1 h at room temperature followed by addition of ECL (PerkinElmer, NEL105001).

### Growth curves and drug responses
Cell growth curve assays were performed for the indicated periods of time using the Sartorius Incucyte imaging system. Exponentially growing cells were seeded into 24-well plates at 1000 cells per well for MEFs. Four images were collected per well at each time point and confluence measurements averaged. A minimum of three biological replicates were performed and the mean confluence was presented with SEM for all replicates. Colony formation assays were performed in six-well plates with 300 MEF cells per well or 500 MDA-MB-436 cells per well in media with the indicated concentrations of ART558 or rucaparib. DMSO concentrations were equalized for each condition. Cells were fixed with 4:1 methanol:acetic acid after 2 weeks and then stained with crystal violet. Colonies were then counted and IC50 was calculated using results from three or more independent experiments. Response curves are shown as mean colony formation ± S.E.M.

### Karyotypic and M-FISH Analysis
Preparation of mouse metaphases and G-banding were carried out as previously described[50]. Cells in logarithmic growth phase were treated with 0.03 μg/ml Colcemid, trypsinized, transferred to hypotonic 0.075 M KCl for 15 min at 37 degrees then fixed in 3:1 methanol: acetic acid. The guidelines used for karyotypic designations and chromosome breakpoint determination of murine metaphase chromosomes are located at http://www.pathology.washington.edu/research/cytopages/idiograms/mouse/. For mFISH analysis, metaphase preparations were hybridized using a 21Xmouse multicolor FISH probe for mouse chromosomes mFISH kit (MetaSystems, Heidelberg, Germany). The probe consists of 21 painting probes specific for the 21 different mouse chromosomes, labeled with different fluorochromes, and the excitation/emission spectra are comparable to aqua, green, orange, red, and near-infrared fluorochromes. Image capturing and processing were performed with a Zeiss AxioImager Z2 fluorescence microscope, with single bandpass filters (Chroma Technology)

appropriate for each fluorochrome and an Isis/mFISH image analysis system (MetaSystems).

## Digital droplet PCR reporter assays

HR and TMEJ were measured by ddPCR assay as previously described[51]. In brief, Cas9 and *Rosa26* targeted sgRNA were electroporated then repair products were amplified across the Cas9-induced break site in the *Rosa26* locus. HR was measured through the incorporation of a homologous repair template using forward primer CCGCCCA-TAGTACTCTGGAG and reverse AGAAAACTGGCCCTTGCCATT. The repair signal was normalized to 100 copies of genomic DNA using a control ddPCR reaction on chromosome 6. HR assay quantifications are presented as a percentage genomic DNA copies, determined by the chromosome 6 control reaction. TMEJ values were determined by summing the quantifications of three validated TMEJ repair products (del23bp, del39bp, del95bp) and normalizing them to the total TMEJ repair products obtained in wild-type cells. The primers for each TMEJ product are as follows: 23bp forward TTTAAGCCTGCCCAGATC, 39bp forward TTTAAGCCTGCCCAGATC, 95bp forward GCCCACACACC AGTCC, and reverse TCAGTTGGGCTGTTTTGGAG.

## END-seq resection measurements

To measure DNA end resection by END-seq, cells were transduced with a retroviral vector encoding doxycycline-inducible expression of estrogen receptor fused nuclease AsiSI (provided by Andre Nussenzweig). Single-cell clones were isolated and selected based on their doxycycline-induced expression levels and nuclear translocation by adding tamox-ifen to the media. END-seq was performed as described previously[52]. In brief, exponentially growing cells were treated for 19 h with 4 mg/ml doxycycline to induce AsiSI expression, followed by the addition of 1 µM tamoxifen for 5 h to stimulate AsiSI nuclear localization. DNA ends were processed, biotinylated adapters ligated, and breaks isolated by strep-tavidin capture followed by PCR amplification and paired-end sequencing using an Illumina NextSeq 2000 and a P2 100 cycle flow cell. Reads were trimmed (Trimmomatic v0.36), PCR duplicates removed (Picard v2.27.5), and mapped to the mouse GRCm38/mm10 genome with BWA v0.7.17-r1188. Subsequent files were generated with SAMtools v1.13 and BEDtools v2.27.1. Normalized read counts were binned for the 20 kb upstream and downstream of each AsiSI genomic location, identified as 5'-GCGATCGC-3' sequences within the mouse genome. Cut AsiSI locations were determined based on peak detection and ordered by resection distance calculated from peak width. Binned read counts are plotted for each cut AsiSI location and shown as heatmaps, centered on the AsiSI genomic site. Resection quantifications are shown for the top 20 resected sites for each sample. END-seq data are available at the GEO repository under accession number GSE244865.

## Live cell imaging

An H2B-mCherry construct was generated by cloning H2B from Addgene plasmid #113086 (gift of Janet Rossant) in an pENTR1A plasmid containing mCherry then transferred to the pLX304 expression plasmid using LR clonase (Invitrogen, 11791020) according to the manufacturer's instructions. MEFs were transduced with lentivirus containing the H2B-mCherry plasmid and selected with Blasticidin. Cells were seeded into Petri dishes with coverslip bottoms (Ibidi, 8128-200) and imaged with a Leica STELLARIS 5 confocal microscope with temperature and carbon dioxide regulation. Three fields of view were imaged per condition for $n = 3$ biological replicates. Maximum projection images were generated and a minimum of 10 mitotic cells were tracked manually per replicate. Abnormal versus normal mitoses were counted and the type of abnormality observed (lagging chromosomes, anaphase bridges, multipolar spindles) was quantified.

The GFP-MDC1 and RPA2-mCherry plasmids were constructed from Addgene plasmids #26427 (gift of Eric Campeau) and #102613 (gift of Marc Wold) and transduced into MEFs as described above. Cells

were additionally sorted for dual expression of GFP and mCherry using a BD FACSAria II system. For GFP-MDC1 and RPA2-mCherry imaging, cells were preincubated with 2 mM thymidine to synchronize in S phase, released for 2 h to allow for replication restart, then treated with DMSO or 10 µM ART558, and imaged. Adjacent fields of view (×20) were collected and stitched together using a Leica STELLARIS 5 con-focal microscope and software. Maximum intensity projections were generated and images of individual nuclei were generated for foci-positive nuclei. A total of 15 foci containing nuclei across $n = 2–3$ replicates were quantified for the number of foci per nucleus over time. Foci were counted using an ImageJ (v.2.1.0/1.53c) macro[6]. Foci identification was performed using the ImageJ background subtraction, contrast, and Gaussian blur functions followed by thresholding to generate a foci mask image. The threshold values were applied con-sistently within each experiment and manually confirmed. The ImageJ analyze particles function was then used to quantify foci for each nucleus from the foci mask image.

## Fixed-cell immunofluorescence

Immunofluorescence foci formation assays were conducted 6–8 h after exposing cells to 2 or 10 Gy γ-irradiation as indicated in figure legends. For Rpa and Rad51 foci formation experiments, cells were pre-extracted on ice for 5 min with cold cytoskeleton buffer (10 mmol/L PIPES pH 6.8, 100 mmol/L NaCl, 300 mmol/L sucrose, 3 mmol/L MgCl2, 1 mmol/L EGTA, 0.5% Triton X-100) followed by 5 min with cytoskeleton stripping buffer (10 mmol/L Tris-HCl pH 7.4, 10 mmol/L NaCl, 3 mmol/L MgCl2, 1% Tween 20 (v/v), 0.5% sodium deoxycholate). Cells were fixed at room temperature with 4% paraformaldehyde for 10 min, washed with PBS, and permeabilized with 1% Triton X-100 in PBS for 10 min at room temperature. The following primary antibodies were incubated overnight at 4 degrees in 5% goat serum in PBS: γH2ax (Millipore Sigma, 05-636, 1:1000), Rad51 (Abcam, ab133534, 1:50000), Rpa32 (Cell Signaling, 2208, 1:1000 for MEFs or Millipore Sigma, NA18, 1:1000 for human cells). Z-stack images were collected on a Leica STELLARIS 5 confocal microscope and maximum intensity projections were generated for at least five fields of view per independent repli-cate. A minimum of 3 biological replicates were performed for each condition. Foci were quantified using an ImageJ (v.2.1.0/1.53c) macro as described above and cells with 5 or more foci counted as positive[6].

## Translocation analysis

Translocation assays were adapted from a previous study[21]. A single cell *Brca1*$^{CC/CC}$ clone expressing doxycycline-inducible Cas9 was iso-lated and expanded following transduction with lentivirus containing pCW-Cas9 plasmid (Addgene plasmid #50661, gift from Eric Lander and David Sabatini). The following single guide RNA sequences tar-geting *Rosa26* and *H3f3b* were cloned into pLenti-F2 and lentivirus generated: ACTCCAGTCTTTCTAGAAGA and GTTGGCTCGCCGGA-TACGGG. For transduction with sgRNA plasmid, 500 cells per well were plated in 96 well plates and the following day were transduced with virus, 1× polybrene (Santa Cruz Biotechnology, sc-134220), and 4 µg/ml doxycycline in the presence of DMSO or 10 µM ART558. Negative controls were generated by excluding sgRNA lentivirus. DNA was collected 96 h after transduction by replacing media with 60 µl of 50 mM NaOH and heated to 95 degrees for 30 min followed by the addition of 15 µl 1 M Tris-HCl pH 8.0 from 5 wells per condition. Cutting at both loci was assessed using the following primers and Sanger sequencing: RosaF GCGGGAGAAATGGATATGAA, RosaR GCACGTT TCCGACTTGAGTT, H3f3bF TTGACGCCTTCCTTCTTCTG, H3f3bR AAC CTTTGAAAAAGCCCACA.

Amplification of a Rosa-H3f3B translocation product was per-formed by PCR using the H3f3bF and RosaR primers by a nested PCR reaction using the following primers: H3f3bNF CTGCCATTCCAGA-GATTGGT and Rosa26NR TCCCAAAGTCGCTCTGAGTT. The first round of PCR was performed with 500 ng genomic DNA for each well

and 4 µl product for the nested reaction, each using 60 degrees annealing temperature. The nested PCR reactions were run on an agarose gel with expected translocation junctions of ~500 bp. Wells containing amplified translocation products were counted and product size was estimated. Technical replicates were performed for each well using 500 ng genomic DNA inputs. Translocation frequency was determined from the total number of translocation-positive wells across both technical replicates and shown for each $n = 4$ biological replicates. For breakpoint sequence analysis, PCR products from individual wells were sequenced by Sanger sequencing and deconvolved using DECODR v3.0 software (decodr.org)[53]. If multiple Sanger sequence traces were detected and could not be reliably deconvolved, those sequences were excluded from further analyses. Of note, ART558 treatment resulted in frequent deletions of varying sizes visible on the agarose gels leading to fewer sequences available for breakpoint analyses. Sequences were aligned to the expected translocation junction and assessed for microhomology, insertions, and deletions at each nucleotide position along the expected product. Data are presented as microhomology or deletion frequencies at each position, centered at the breakpoint. Additionally, total deletion sizes were determined for each sequence and shown separately.

### RNA interference
Exponentially growing cells were transfected with 20 nM pooled *Brca1* targeted siRNA (equal parts of the following siRNAs from Dharmacon: J-040545-05-0002, J-040545-06-0002, J-040545-07-0002, J-040545-08-0002) or 20 nM scrambled (Qiagen, SI03650318) siRNA using the Lipofectamine RNAiMAX transfection reagent (Invitrogen, 13778150) per manufacturer instructions. Immunofluorescence and western blotting assays were conducted after 72 h culture in media containing siRNA. Colony assays were performed in the continued presence of siRNA for their duration.

### qRT-PCR analysis
RNA was extracted from cell pellets following manufacturer instructions (Qiagen, 74104). Reverse transcription with M-MLV (Thermo Fisher, 28025013) and PCR using the PowerUp SYBR Green Master Mix (Thermo Fisher, A25743) were performed using conditions recommended by the manufacturer and the following primers: Polq-F CTGTATGCCTCTGGCTTTCTC, Polq-R CTTCAGCTGCTTCCTCTTCTT. *Polq* expression was normalized to *36b4* (*RplpO*) expression, determined using the following primers: F GCTCCAAGCAGATGCAGCA, R CCGGATGTGAGGCAGCAG.

### Cancer cell lines
MDA-MB-436 cells were cultured in RPMI with 10% fetal bovine serum and 1× penicillin/streptomycin. A doxycycline-inducible Cas9 expression single cell clone was generated following transduction with lentivirus containing a modified pCW-Cas9 plasmid (Addgene plasmid#52961, gift of Feng Zhang) where the puromycin resistance gene was replaced with BFP. Single guide RNA targeting BFP (control), PALB2, or BRCA2 with the following sequences were inserted into a sgRNA expression plasmid with puromycin selection: BFP: ACGCCCCCGTCTTCGTATG, BRCA2: GCAGGTTCAGAATTATAGGG, PALB2: AGCTGAGGGGCTTCCCGGG. Single-cell clones were isolated and sequenced using the following primers and knockout confirmed by Western blotting: BRCA2-forward: AACAAAAGTAATCCATAGTCAAGAT, BRCA2-reverse: GTTTGCCTAAATTCCTAGTTTGT, PALB2-forward: GTGGCCCACTGGGAC, PALB2-reverse: AGAGGAGGGGGTGGTCAGAT. Ectopic BRCA1-Δ800 was added back using pLenti-BRCA-M800-IRES-HygroR and selected with 100 µg/ml Hygromycin. BRCA1-Δ800 construct lacks 800 amino acids from within exon 11 but retains BRCA1 resection and RAD51 loading functions[35]. No endogenous BRCA1 was detectable by Western blot in the MDA-MB-436 cells which harbor a 5396 + 1 G > A mutation.

### Statistics and reproducibility
Statistical tests, number of replicates, and *p* values are described in the figure legends. GraphPad Prism Version 9.4.0 was used for unpaired, two-tailed *t* tests and comparisons indicated in the figures. R and RStudio v1.3.1093 were used for chi-square goodness of fit tests comparing expected and observed genotypes.

### Reporting summary
Further information on research design is available in the Nature Portfolio Reporting Summary linked to this article.

## Data availability
All relevant data and unique materials are available from the authors upon request. The raw data generated in this study are provided in the Source Data file. The END-seq data generated in this study have been deposited in NCBI's Gene Expression Omnibus and are accessible through GEO Series accession number GSE244865. Source data are provided in this paper.

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

## Acknowledgements

This work was supported by US National Institutes of Health (NIH) Grants to N.J. P30CA006927, R01CA214799, R01CA255360, and R01GM135293. J.J.K. was supported by an American Cancer Society—Tri State CEOs Against Cancer Postdoctoral Fellowship, PF-19-097–01–DMC, Ovarian Cancer Research Alliance, and Phil and Judy Messing grant 597484. NCI P01 CA247773 and University Cancer Research Fund (GPG). D.J.G. is supported by T32GM142606. We are grateful to the FCCC Laboratory Animal, Cell Culture, Biological Imaging, and Cell Sorting facilities. We thank Dr. Kathy Cai and the FCCC histopathology service for pathological analyses of mouse organs, and Dr. Beth Handorf for guidance with statistical analyses. We thank Artios Pharma for supplying ART558.

## Author contributions

J.J.K., D.G., I.C., Y.W., W.F., D.A.S., P.P., Z.L., R.N.D., R.B., A.J.B., A.M.B., J.C., W.T.Y., J.N., designed and performed experiments. J.J.K., R.P.,

G.P.G., JR.T., and N.J. analyzed and interpreted data. J.J.K. and N.J. wrote the manuscript and supervised the project.

## Competing interests

G.P.G. has received research funding from Breakpoint Therapeutics and Merck. R.T.P. is a cofounder and chief scientific officer of Recombination Therapeutics, LLC. The remaining authors declare no competing interests.
