## [Peer Review File · Nature Communications]

Genetic separation of Brca1 functions reveal mutation-dependent Pol θ vulnerabilitiesREVIEWER COMMENTS

Reviewer #1 (Remarks to the Author):

The synthetic lethal relationship between Pol θ and proteins required for homologous recombination (HR) such as BRCA1 and BRCA2 is well known. HR deficient cells also exhibit synthetic lethality to PARP inhibitors. BRCA1/53BP1 double mutant cells are resistant to PARPi due to defect in end resection and restoration of HR. Interestingly, 53BP1 defective cells are sensitive to Pol θ inhibitors. Given the impact of end resection on sensitivity to Pol θ inhibitor ART558, in this manuscript, the authors have examined the impact of two different Brca1 mutant alleles, Δ 11 allele (lacking exon 11) and the cc allele, which is defective in interaction with PALB2 on the synthetic lethality induced by DNA Pol θ deficiency or inhibition, in the presence or absence of 53BP1 in cells as well as mice. The results show that viability of Brca1 mutant cells and mice on a Pol θ deficient background varies between the two allele. However, in the absence of 53BP1 and Pol θ , cells and mice with either of the two alleles are not viable. The authors show that the differential response of the two Brca1 alleles to Pol θ inhibitor is associated the ability of mutant BRCA1 to promote end resection and load RAD51 via interaction with PALB2. Ability to perform end resection (in the case of cc mutant) promotes Pol θ mediated end joining rendering them more sensitive to ART558. However, because this mutant is defective in RAD51, it retains sensitivity to PARPi. In contrast, cells expressing Δ 11 allele is defective in end resection and exhibits less sensitivity to ART955. Since PALB2 and BRCA2 mutant cells have normal end resection, they too were shown to exhibit high sensitivity to ART558.

The clinical implications of the findings reported in the manuscript are significant. Not all mutant alleles of BRCA1 are expected to exhibit a similar response to Pol θ inhibitors. Therefore stratification of patients with different BRCA1 mutations should be an important consideration in the clinic. The experiments are performed very well using proper controls and statistical analysis. The manuscript is clearly written. Because of the use of at least two different classes of Pol θ inhibitors, such as Novobiocin and ART558 and its derivatives in clinical trials. While both Novobiocin and ART558 are Pol θ inhibitors, their mode of action seems to be different. Therefore, it will be important to test the effect of Novobiocin as shown for ART558 in Figure 3.

Reviewer #2 (Remarks to the Author):

The primary purpose of your review is to provide feedback on the soundness of the research reported. This will help authors to improve their manuscript and editors to reach a decision. When composing your report, the following questions might assist you in writing a well-justified review, but please feel free to raise any further questions and concerns about the paper.

- What are the noteworthy results?

BRCA1 and BRCA2 mutations are not equivalent in terms of their synthetic lethal interactions with POLQ. BRCA1 is associated with stimulating resection whereas BRCA2 is involved in RAD51 loading and stabilizing nucleoprotein filaments. BRCA1, therefore, would be expected to have defects in resection whereas BRCA2 should be resection competent. TMEJ likely relies on resection so the use of two separation-of-function alleles in BRCA1, delExon11 (resection defective) and CC (resection competent but unable to bind PALB2), is highly informative in terms of their genetic interactions with POLQ loss.

This study gets at the molecular mechanism underlying the differences between the delExon11 and CC mutant BRCA1 alleles in terms of their genetic interactions with POLQ loss and treatment with POLQi.

The study contains a rich source of information for understanding the genetic interplay and pharmacological inhibition of POLQ and PARP in terms of BRCA1, BRCA2, and PALB2 in relation to 53BP1. Notable results include: (1) loss of POLQ reverses 53BP1^{-/-} rescue of BRCA1 mutant alleles CC and delExon11. In seminal mouse studies by Bunting et al, 53BP1^{-/-} rescues BRCA1 loss, however, POLQ loss prevents this rescue suggesting TMEJ is required for viability/survival. (2) Polq^{-/-} cells could not be derived (CRISPR/Cas9 KO) in 53BP1 ko background with either CC or del11 brca1 alleles, (3) interestingly, brca1^{-/-}/polq^{-/-} cells were viable albeit they grew slowly and were chromosomally unstable, (4) brca1 mutant cells that could resect, CC mutant, were more dependent on POLQ for viability than the delExon11 mutant, (5) loss of 53BP1 in combination with brca1 CC mutation resulted in mild (3-fold) increase in PARPi resistance (probably because CC mutant is already resection competent, i.e is not blocked by 53BP1), (6) loss of 53BP1 in combination with brca1 delExon11 mutation resulted in dramatic (130-fold) increase in PARPi resistance (delExon11 is resection defective so removing 53BP1 allowed resection to move forward and this mutant can bind PALB2 and assist in RAD5 loading), (7) CC mutant cells exhibited high levels of RPA foci upon POLQi treatment, (8) PALB2 and BRCA2 mutant cells (resection competent) were sensitive to POLQ, however, functional BRCA1 (implying functional resection) further sensitized to POLQi. Thus, resection is critical for POLQi response in HRD cells.

- Will the work be of significance to the field and related fields? How does it compare to the established literature? If the work is not original, please provide relevant references.

This is an excellent and thorough workup of two important brca1 mutant alleles. From a clinical standpoint, brca1 frameshift mutations in Exon 11 account for 1/3 of all pathogenic mutations so the work is highly significant as this allele is important to study and understand.

- Does the work support the conclusions and claims, or is additional evidence needed?

The conclusions and claims are supported by the experimental data.

- Are there any flaws in the data analysis, interpretation and conclusions? Do these prohibit publication or require revision?

Minor, please see comments below. Minor revision required.

- Is the methodology sound? Does the work meet the expected standards in your field?

The rigor is high, multiple isogenic cell lines used, in vivo mouse models are used. Orthogonal approaches are used to verify results. The work is high quality.

- Is there enough detail provided in the methods for the work to be reproduced? yes

Where applicable, reporting summaries are requested from the authors to improve the transparency and reproducibility of published results. We hope the file, if included, will aid in your evaluation of the paper as they contain key information pertaining to study design and analysis.

Review of "Genetic separation of Brca1 functions reveal pre-mitotic RPA as a driver of Polq addiction" by Kraiss et al.

The paper by Kraiss et al. is clearly written and the experiments are well designed and executed. The figures are straightforward with some suggestions noted below (e.g the schematics/cartoon models are somewhat confusing and could be misconstrued in Fig 1A, 4H, & 6F). The approaches are well constructed and the data is of high quality. There is a lot of data packed into this paper but the

methodology and rationale are well laid out. The use of cell lines and mouse crosses is complementary and a strength justifying the conclusions of the authors. Some of the most interesting results highlight the different way in which BRCA1 deficient cells respond to POLQ deletion, how the separation-of-function *brca1* alleles (CC and del11) respond differently to PARPi and POLQi, and the differential response to POLQi imparted by BRCA1 status in PALB2 and BRCA2 mutant cells.

This paper is important as it helps the reader conceptually understand the significance of complex relationships between HR deficiencies, how resection is modulated, and consequences of POLQ loss in those two backgrounds. The two BRCA1 alleles, CC and del11, provide nice separation-of-function to understand the consequences of POLQ loss in those two backgrounds layered on top of 53BP1 dependence. The isogenic comparisons between the double and triple genetic modifications are very helpful to understand the repair pathways and how these genotypes dictate viability and response to PARPi and POLQi.

My only major critique is that the title does not seem to convey what the paper is really about and I would suggest a title that reflects more broadly on the characterization of the two *brca1* separation-of-function alleles CC and del11 rather than the one phenotype explored in Fig 5 showing increased RPA foci/levels in the CC vs del11 mutant. Perhaps something like "Distinct BRCA1 mutant alleles give rise to differential POLQ dependencies" or add some language in the title that relays to the reader the importance of these two *brca1* alleles in terms of their ability to facilitate resection which seems critical for the POLQ dependence.

BRCA1/2 are thought to be synthetic lethal with POLQ, however, this study demonstrates that this is not always the case and is dependent upon the type of mutation. The authors use a *brca1* "del11" mutant (Exon11 is deleted) that is defective for resection as this allele can't counteract 53BP1. The other *brca1* allele "CC" is resection competent but can't bind PALB2 and cells expressing this mutant exhibit defective RAD51 loading (e.g. lack of RAD51 foci). The authors complete an exhaustive workup of these two alleles in both multiple cell lines and in mouse crosses. They combine the two *brca1* alleles (CC and del11) with knockouts or knockdowns of POLQ and 53BP1 to determine their genetic dependencies. Interestingly, in mouse crosses, they find that loss of POLQ reverses the loss of 53BP1 rescue of both *brca1* mutant alleles. At a low frequency, they were able to obtain *brca1* mutant allele/*polq* knockout cell lines (in 53BP1+/+ cells only) and showed that while CC exhibited more severe phenotypes, both *brca1* mutant allele cell lines suffered chromosomal instability highlighted by marker chromosomes with segments derived from up to 5 different chromosomes. In response to PARPi treatment, the del11 cells were highly sensitive to rucaparib but became 130-fold more resistant upon 53BP1 loss. The CC cells were also sensitive to PARPi which was only moderately changed (3-fold) upon 53BP1 loss perhaps reflecting a difference in the ability of this allele to resect versus del11 which is resection defective. Regardless of 53BP1 status, CC cells were more sensitive to POLQi in comparison to del11 cells, however, upon 53BP1 loss, both CC and del11 cells became about 2.5-fold more sensitive to POLQi. These results have clear implications for how patient tumors may or may not benefit from PARPi and/or POLQi treatment based upon 53BP1 status of the tumor and the particular BRCA/HR gene mutation.

In Figure 4, the authors measure HR, TMEJ, and resection comparing WT to CC and del11 with and without 53BP1. HR was measured by digital droplet (dd)PCR following Cas9 induced DSB at the Rosa26 locus and CC and del11 HR levels were low compared to WT as expected. Upon 53BP1 loss, del11 HR activity increased as resection was now no longer impeded by 53BP1 and likely the increase was driven by this allele's ability to bind PALB2 and stimulate RAD51 loading at resected DSBs. TMEJ was measured by ddPCR for 3 distinct repair products and increased dramatically upon 53BP1 loss for WT and the CC allele, but less for the del11 allele. END-seq was then used to quantify resection (at AsiSI-induced DSBs) confirming resection defect in del11 cells and resection was increased across all genotypes upon 53BP1 loss as anticipated. The authors then use a complementary approach to measure resection by radiation induced RPA foci and the WT, CC, and del11 displayed similar trends to the END-seq with CC levels similar to WT but del11 RPA foci were dramatically lower. 53BP1 loss

increased RPA foci across all genotypes. RAD51 foci were found to be marginal in both CC and del11, however, upon 53BP1 loss, del11 RAD51 foci increased greatly consistent with this allele's ability to bind PALB2 and assist in RAD51 loading. To confirm that the del11 allele was responsible for RAD51 foci in these cells, the authors knocked down the del11 protein using siRNA and RAD51 foci plummeted. Furthermore, they found that depleting the brca1 del11 protein further sensitized the cells to POLQi.

In Figure 5A, the authors find that gamma-H2AX induction was restricted to CC cells in response to POLQi. These results are consistent with CC cells being more sensitive to POLQi shown in Fig. 3B. The authors then demonstrate that CC cells contain more mitotic abnormalities, micronuclei, and lagging chromosomes after POLQi treatment than WT and del11 cells. Synchronized cells were then used to track GFP-MDC1 and RPA-mCherry as markers of DNA breaks and ssDNA, respectively following POLQi treatment. MDC1 levels (DNA breaks) were consistently higher in CC cells compared to del11 cells pre-, during, and post-mitosis whereas RPA levels (indicator of resection) were significantly higher in CC cells even during mitosis and in the daughter cells. The authors speculate that CC cells, which are resection competent, engage TMEJ but upon POLQi, ssDNA overhangs coated with RPA persist. Interestingly, CC cells did not show an increase in translocation frequency in response to POLQi at a Cas9-induced DSB at the Rosa26 locus, in fact, a slight decrease was observed. Larger deletions were observed around the DSB in CC cells upon POLQi treatment possibly indicating aberrant SSA or NHEJ repair.

Finally, in Fig.6, the authors utilize a different cell line, human MDA-MB-436 cells, which are BRCA1 mutant. They knock out PALB2 and/or complement with the BRCA1 cDNA in these cells. They observe that radiation induced RPA foci are dependent upon BRCA1 but not PALB2, and that RAD51 foci depend on PALB2. Cells that had lost BRCA1 or PALB2 were sensitive to PARPi whereas, strikingly, PALB2^{-/-} cells with functional BRCA1 are much more sensitive to POLQi than PALB2^{-/-}BRCA1^{-/-} cells. Similar results were found for BRCA2^{-/-}BRCA1^{-/-} cells being less sensitive to POLQi than the same cells complemented with BRCA1. Thus, these results suggest that BRCA1-driven resection is an important indicator of sensitization to POLQi. This is another important result with clinical implications for how patients should be stratified for PARPi and/or POLQi treatment.

Major details:

1. Change the title! - very little in text about POLQ addiction and pre-mitotic RPA. Title should convey bulk of the work. Also, title as written seems bit confusing.
2. Figure 2 could go into Supplementary(Extended) data.

Minor details:

1. Introduction – provide one sentence explaining how BRCA1 counteracts 53BP1-shieldin etc...
2. P4 – “genotypes were readily ‘obtained’?” what is the source of these SV40 immortalized cells?
3. P5 top – “likely producing only a small non-functional peptide”. Did the authors check for POLQ protein by western/Mass Spec?
4. Fig 1A - The cartoon/model schematic is confusing, it looks like BRCA2delExon11 inhibits 53BP1 & Shieldin etc...but the data & text suggests this allele is defective for resection meaning it can't overcome blockade by 53BP1. Same for CC allele, maybe delete the red inhibition line and depict that the two alleles are defective in some other manner.
5. Fig 1D, either in text or discussion, would be good to discuss that del11 cells can't resect so they don't rely on POLQ (TMEJ is low, supported by data in Fig 4B) perhaps explaining why polq^{-/-} 50% is similar to 50% for WT. While CC cells resect (but can't bind PALB2) and rely on POLQ for viability (TMEJ is higher).
6. Are WT cells that lose POLQ sensitive to PARPi? Precedence for this in literature? Ceccaldi et al

Nature 2015 show MDA-MB-436 Pol Theta kd same PARPI sensitivity as shScr. How about the cells used in this study?

7. P5 bottom – “Taken together, the type of Brca1 mutation strongly influenced Polq-synthetic lethality, whereas when combined with 53bp1 deficiency, both Brca1 mutations required Polq for cellular and embryonic viability.” This sentence is bit confusing. The paragraph is about results in mouse crosses where it doesn’t matter which brca1 allele is used (CC or del11) both are lethal with polq-/. Is this sentence referring to Fig 1D, where in 53BP1+/+ cell line, del11 colonies are more than CC? Please clarify in text.

8. P6 top – “Fig. S2b” should be S2c.

9. P7 “..suggesting HR is restored to varying degrees, and examined in greater detail in the next section” Confusing, please clarify and where is this examined in next section?

10. P7 – “showing exquisite sensitivity” – 2.3 vs 2.6 fold doesn’t seem like exquisite is right word here.

11. P7 – “reproducing” should be reproduced

12. P8 top – “increased sensitivity to ART558”. Maybe add to this sentence, “ and yet, increased resistance to PARPi.

13. P9 – explain one sentence how H2B-mCherry measures mitotic abnormalities.

14. P11- Explain what the BRCA1 mutation is in MDA-MB-436- completely null at protein level?

15. Fig. 2A – are these MEFs from Fig. 1D? Please label cell types in figures.

16. Fig. 3 – please put IC50 label in figure next to Rucarib and ART558. Might help to notate fold change in resistance/sensitivity next to #'s

17. Fig 4A – why didn’t HR increase in WT when 53BP1 knocked out? (went down??)

18. Fig. 4 F – draw an arrow indicating del11 protein. Fig. 4F-G Indicate genotype of cells here – del11/53BP1-/-

19. Fig.5 – Are these cells 53BP1-/-? Fig. 5C Label the images with MDC1 and RPA. Is Pre-M same as P below? And Post-M is D? If so, label them the same. Fig. 5D – Label which cells are used here.

20. Fig. 6F – scheme/model is not clear

21. Fig S1D – shouldn’t 53BP1 be -/- not +/+?

Concepts to think about:

Fig. 3 – Is it surprising that del11 allele didn’t become more sensitive to POLQi (2.3-fold) after 53BP1 deletion? Resection would be upregulated but TMEJ inhibited. Whereas CC is already resection competent so might be expected that 53BP1 loss would not change POLQi sensitivity that much? And is it surprising that del11 is sensitive to POLQi at all given that it is defective for resection? Might help to discuss difference between genetic POLQ knockout and pharmacological inhibition of POLQ in text.

Reviewer #3 (Remarks to the Author):

In Nature Communications submission NCOMMS-23-35790-T, Kraiss, Johnson and co-authors perform genetic experiments with Brca1 separation-of-function mutant mice and derived cell-lines, to determine the genetic and mechanistic determinants of Pol-theta (POLQ) loss/inhibition-dependent synthetic sick relationship with BRCA1-mutation associated HR-deficiency. Contrasting the genetic synergy between POLQ-loss/inhibition in resection-deficient Brca1 Δ 11/ Δ 11 and Rad51 recombinase recruitment/loading -deficient Brca1CC/CC, across a series of cellular readouts of cell survival, growth, DNA repair and chromosomal stability, they make a compelling case for DNA end resection proficiency being the critical determinant of PolQi sensitivity in HR-deficient cells. A major strength of this work is the genetics and data quality, which resolves each phenotype into clear-cut, unambiguous results with the dynamic range to clearly contrast POLQ interactions within distinct yet related backgrounds. Many of the major conclusions are supported by state-of-the-art methodologies, e.g. END-seq to measure

resection (complementing RPA IRIF data), and ddPCR method to give absolute measures of pathway usage between the different backgrounds. The PALB2 KO experiment in Figure 6 is also a neat “killer experiment” that successfully attests their major conclusion regarding BRCA1-dependent resection control.

While the work builds on previous findings regarding POLQⁱ/loss vs HRD synthetic lethality, and the role of resection control (ie via 53BP1 vs resection enzymes), this is a first-time from a BRCA1-mutation centric perspective. This is one of the most clinically-relevant indications for potential POLQⁱ therapies, and as such the conclusion can be deemed of high scientific interest and medical importance. In my opinion this is a compelling, fascinating and highly convincing study that requires minimal alterations. It should be considered a high-impact/priority manuscript for publication in Nature Communications.

Comments:

1. Synthetic lethality is used quite a lot in the manuscript, and in some occasions, a more nuanced statement (synthetic sick??) might more appropriately describe the genetic interaction.
2. Consider revising Fig1A as it's confusing – from the depiction on the right (a variation on a similar schematic from Nacson et al., Mol Cell 2021), it looks like the Brca1 del11 protein is actually able to inhibit 53BP1 (when they speculate it's the opposite); likewise BRCA1 deltaCC looks like it's an active inhibitor of PALB2-dependent RAD51 loading (which is the wrong way round).
3. The statement “Translocation frequency did not increase with ART558, rather, a decrease was observed, although not reaching significance (Fig. 5d, S3c). As expected, ART558 resulted in fewer junctions containing microhomology (Fig. 5e, S3d)” is misleading. If the findings are not statistically significant, the noted effect might simply be down to chance.
4. Figures S2C and S2D have been incorrectly cited.
5. Scale bars in Figure 2B could be included to facilitate accurate comparisons of the chromosomes.

REVIEWER COMMENTS

We appreciate the positive comments from all three Reviewers. Below, we provide a response to general comments, as well as point-by-point responses to specific issues.

Reviewer #1 (Remarks to the Author):

The synthetic lethal relationship between Pol θ and proteins required for homologous recombination (HR) such as BRCA1 and BRCA2 is well known. HR deficient cells also exhibit synthetic lethality to PARP inhibitors. BRCA1/53BP1 double mutant cells are resistant to PARPi due to defect in end resection and restoration of HR. Interestingly, 53BP1 defective cells are sensitive to Pol θ inhibitors. Given the impact of end resection on sensitivity to Pol θ inhibitor ART558, in this manuscript, the authors have examined the impact of two different Brca1 mutant alleles, Δ 11 allele (lacking exon 11) and the cc allele, which is defective in interaction with PALB2 on the synthetic lethality induced by DNA Pol θ deficiency or inhibition, in the presence or absence of 53BP in cells as well as mice. The results show that viability of Brca1 mutant cells and mice on a Pol θ deficient background varies between the two allele. However, in the absence of 53BP1 and Pol θ , cells and mice with either of the two alleles are not viable. The authors show that the differential response of the two Brca1 alleles to Pol θ inhibitor is associated the ability of mutant BRCA1 to promote end resection and load RAD51 via interaction with PALB2. Ability to perform end resection (in the case of cc mutant) promotes Pol θ mediated end joining rendering them more sensitive to ART558. However, because this mutant is defective in RAD51, it retains sensitivity to PARPi. In contrast, cells expressing Δ 11 allele is defective in end resection and exhibits less sensitivity to ART955. Since PALB2 and BRCA2 mutant cells have normal end resection, they too were shown to exhibit high sensitivity to ART558.

The clinical implications of the findings reported in the manuscript are significant. Not all mutant alleles of BRCA1 are expected to exhibit a similar response to Pol θ inhibitors. Therefore stratification of patients with different BRCA1 mutations should be an important consideration in the clinic. The experiments are performed very well using proper controls and statistical analysis. The manuscript is clearly written. Because of the use of at least two different classes of Pol θ inhibitors, such as Novobiocin and ATR558 and its derivatives in clinical trials. While both Novobiocin and ART558 are Pol θ inhibitors, their mode of action seems to be different. Therefore, it will be important to test the effect of Novobiocin as shown for ART558 in Figure 3.

We thank the Reviewer for emphasizing the clinical importance of this work for patients receiving Pol θ inhibitors. Given the focus of our manuscript is on the *genetic relationship* between HR gene mutations and Pol θ dependencies, we primarily use ART558 as a tool compound as it is well known to be a specific and on-target inhibitor of Pol θ . Indeed, novobiocin is also a potent topoisomerase and DNA gyrase inhibitor (PMID: 794878) and therefore, could complicate interpretations of assessing Pol θ dependencies. Furthermore, there are many small molecule Pol θ inhibitors in pre-clinical and clinical development, and by the same rationale, other compounds may need to be tested.

Therefore, we believe testing novobiocin is beyond the scope of the current manuscript. However, we note the reviewers point and mention that it will be interesting to test more Polθ inhibitors in the discussion. See page 14, first paragraph.

Reviewer #2 (Remarks to the Author):

The primary purpose of your review is to provide feedback on the soundness of the research reported. This will help authors to improve their manuscript and editors to reach a decision. When composing your report, the following questions might assist you in writing a well-justified review, but please feel free to raise any further questions and concerns about the paper.

- What are the noteworthy results?

BRCA1 and BRCA2 mutations are not equivalent in terms of their synthetic lethal interactions with POLQ. BRCA1 is associated with stimulating resection whereas BRCA2 is involved in RAD51 loading and stabilizing nucleoprotein filaments. BRCA1, therefore, would be expected to have defects in resection whereas BRCA2 should be resection competent. TMEJ likely relies on resection so the use of two separation-of-function alleles in BRCA1, delExon11 (resection defective) and CC (resection competent but unable to bind PALB2), is highly informative in terms of their genetic interactions with POLQ loss.

This study gets at the molecular mechanism underlying the differences between the delExon11 and CC mutant BRCA1 alleles in terms of their genetic interactions with POLQ loss and treatment with POLQi. The study contains a rich source of information for understanding the genetic interplay and pharmacological inhibition of POLQ and PARP in terms of BRCA1, BRCA2, and PALB2 in relation to 53BP1. Notable results include: (1) loss of POLQ reverses 53BP1^{-/-} rescue of BRCA1 mutant alleles CC and delExon11. In seminal mouse studies by Bunting et al, 53BP1^{-/-} rescues BRCA1 loss, however, POLQ loss prevents this rescue suggesting TMEJ is required for viability/survival. (2) Polq^{-/-} cells could not be derived (CRISPR/Cas9 KO) in 53BP1 ko background with either CC or del11 brca1 alleles, (3) interestingly, brca1^{-/-}/polq^{-/-} cells were viable albeit they grew slowly and were chromosomally unstable, (4) brca1 mutant cells that could resect, CC mutant, were more dependent on POLQ for viability than the delExon11 mutant, (5) loss of 53BP1 in combination with brca1 CC mutation resulted in mild (3-fold) increase in PARPi resistance (probably because CC mutant is already resection competent, i.e is not blocked by 53BP1), (6) loss of 53BP1 in combination with brca1 delExon11 mutation resulted in dramatic (130-fold) increase in PARPi resistance (delExon11 is resection defective so removing 53BP1 allowed resection to move forward and this mutant can bind PALB2 and assist in RAD5 loading), (7) CC mutant cells exhibited high levels of RPA foci upon POLQi treatment, (8) PALB2 and BRCA2 mutant cells (resection competent) were sensitive to POLQ, however, functional BRCA1 (implying functional resection) further sensitized to POLQi. Thus, resection is critical for POLQi response in HRD cells.

- Will the work be of significance to the field and related fields? How does it compare to the established literature? If the work is not original, please provide relevant references.

This is an excellent and thorough workup of two important brca1 mutant alleles. From a clinical standpoint, brca1 frameshift mutations in Exon 11 account for 1/3 of all pathogenic mutations so the work is highly significant as this allele is important to study and understand.

- Does the work support the conclusions and claims, or is additional evidence needed?

The conclusions and claims are supported by the experimental data.

- Are there any flaws in the data analysis, interpretation and conclusions? Do these prohibit publication or require revision?

Minor, please see comments below. Minor revision required.

- Is the methodology sound? Does the work meet the expected standards in your field?

The rigor is high, multiple isogenic cell lines used, in vivo mouse models are used. Orthogonal approaches are used to verify results. The work is high quality.

- Is there enough detail provided in the methods for the work to be reproduced? yes

Where applicable, reporting summaries are requested from the authors to improve the transparency and reproducibility of published results. We hope the file, if included, will aid in your evaluation of the paper as they contain key information pertaining to study design and analysis.

Review of “Genetic separation of Brca1 functions reveal pre-mitotic RPA as a driver of Polq addiction” by Kraiss et al.

The paper by Kraiss et al. is clearly written and the experiments are well designed and executed. The figures are straightforward with some suggestions noted below (e.g the schematics/cartoon models are somewhat confusing and could be misconstrued in Fig 1A, 4H, & 6F). The approaches are well constructed and the data is of high quality. There is a lot of data packed into this paper but the methodology and rationale are well laid out. The use of cell lines and mouse crosses is complementary and a strength justifying the conclusions of the authors. Some of the most interesting results highlight the different way in which BRCA1 deficient cells respond to POLQ deletion, how the separation-of-function brca1 alleles (CC and del11) respond differently to PARPi and POLQi, and the differential response to POLQi imparted by BRCA1 status in PALB2 and BRCA2 mutant cells.

We thank the Reviewer for their positive feedback and have made the recommended changes.

This paper is important as it helps the reader conceptually understand the significance of complex relationships between HR deficiencies, how resection is modulated, and consequences of POLQ loss in those two backgrounds. The two BRCA1 alleles, CC and del11, provide nice separation-of-function to understand the consequences of POLQ loss in those two backgrounds layered on top of 53BP1 dependence. The isogenic comparisons between the double and triple genetic modifications are very helpful to understand the repair pathways and how these genotypes dictate viability and response to PARPi and POLQi.

My only major critique is that the title does not seem to convey what the paper is really about and I would suggest a title that reflects more broadly on the characterization of the two brca1 separation-of-function alleles CC and del11 rather than the one phenotype explored in Fig 5 showing increased RPA foci/levels in the CC vs del11 mutant. Perhaps something like “Distinct BRCA1 mutant alleles give rise to differential POLQ dependencies” or add some language in the title that relays to the reader the importance of these two brca1 alleles in terms of their ability to facilitate resection which seems critical for the POLQ dependence.

We have modified the title according to the Reviewer’s suggestion.

BRCA1/2 are thought to be synthetic lethal with POLQ, however, this study demonstrates that this is not always the case and is dependent upon the type of mutation. The authors use a brca1 “del11” mutant (Exon11 is deleted) that is defective for resection as this allele can’t counteract 53BP1. The other brca1 allele “CC” is resection competent but can’t bind PALB2 and cells expressing this mutant exhibit defective RAD51 loading (e.g. lack of RAD51 foci). The authors complete an exhaustive workup of these two alleles in both multiple cell lines and in mouse crosses. They combine the two brca1 alleles (CC and del11) with knockouts or knockdowns of POLQ and 53BP1 to determine their genetic dependencies. Interestingly, in mouse crosses, they find that loss of POLQ reverses the loss of 53BP1 rescue of both brca1 mutant alleles. At a low frequency, they were able to obtain brca1 mutant allele/polq knockout cell lines (in 53BP1+/+ cells only) and showed that while CC exhibited more severe phenotypes, both brca1 mutant allele cell lines suffered chromosomal instability highlighted by marker chromosomes with segments derived from up to 5 different chromosomes. In response to PARPi treatment, the del11 cells were highly sensitive to rucaparib but became 130-fold more resistant upon 53BP1 loss. The CC cells were also sensitive to PARPi which was only moderately changed (3-fold) upon 53BP1 loss perhaps reflecting a difference in the ability of this allele to resect versus del11 which is resection defective. Regardless of 53BP1 status, CC cells were more sensitive to POLQi in comparison to del11 cells, however, upon 53BP1 loss, both CC and del11 cells became about 2.5-fold more sensitive to POLQi. These results have clear implications for how patient tumors may or may not benefit from PARPi and/or POLQi treatment based upon 53BP1 status of the tumor and the particular BRCA/HR gene mutation.

In Figure 4, the authors measure HR, TMEJ, and resection comparing WT to CC and del11 with and without 53BP1. HR was measured by digital droplet (dd)PCR following Cas9 induced DSB at the Rosa26 locus and CC and del11 HR levels were low compared to WT as expected. Upon

53BP1 loss, del11 HR activity increased as resection was now no longer impeded by 53BP1 and likely the increase was driven by this allele's ability to bind PALB2 and stimulate RAD51 loading at resected DSBs. TMEJ was measured by ddPCR for 3 distinct repair products and increased dramatically upon 53BP1 loss for WT and the CC allele, but less for the del11 allele. END-seq was then used to quantify resection (at AsiSI-induced DSBs) confirming resection defect in del11 cells and resection was increased across all genotypes upon 53BP1 loss as anticipated. The authors then use a complementary approach to measure resection by radiation induced RPA foci and the WT, CC, and del11 displayed similar trends to the END-seq with CC levels similar to WT but del11 RPA foci were dramatically lower. 53BP1 loss increased RPA foci across all genotypes. RAD51 foci were found to be marginal in both CC and del11, however, upon 53BP1 loss, del11 RAD51 foci increased greatly consistent with this allele's ability to bind PALB2 and assist in RAD51 loading. To confirm that the del11 allele was responsible for RAD51 foci in these cells, the authors knocked down the del11 protein using siRNA and RAD51 foci plummeted. Furthermore, they found that depleting the brca1 del11 protein further sensitized the cells to POLQi.

In Figure 5A, the authors find that gamma-H2AX induction was restricted to CC cells in response to POLQi. These results are consistent with CC cells being more sensitive to POLQi shown in Fig. 3B. The authors then demonstrate that CC cells contain more mitotic abnormalities, micronuclei, and lagging chromosomes after POLQi treatment than WT and del11 cells. Synchronized cells were then used to track GFP-MDC1 and RPA-mCherry as markers of DNA breaks and ssDNA, respectively following POLQi treatment. MDC1 levels (DNA breaks) were consistently higher in CC cells compared to del11 cells pre-, during, and post-mitosis whereas RPA levels (indicator of resection) were significantly higher in CC cells even during mitosis and in the daughter cells. The authors speculate that CC cells, which are resection competent, engage TMEJ but upon POLQi, ssDNA overhangs coated with RPA persist. Interestingly, CC cells did not show an increase in translocation frequency in response to POLQi at a Cas9-induced DSB at the Rosa26 locus, in fact, a slight decrease was observed. Larger deletions were observed around the DSB in CC cells upon POLQi treatment possibly indicating aberrant SSA or NHEJ repair.

Finally, in Fig.6, the authors utilize a different cell line, human MDA-MB-436 cells, which are BRCA1 mutant. They knock out PALB2 and/or complement with the BRCA1 cDNA in these cells. They observe that radiation induced RPA foci are dependent upon BRCA1 but not PALB2, and that RAD51 foci depend on PALB2. Cells that had lost BRCA1 or PALB2 were sensitive to PARPi whereas, strikingly, PALB2^{-/-} cells with functional BRCA1 are much more sensitive to POLQi than PALB2^{-/-}BRCA1^{-/-} cells. Similar results were found for BRCA2^{-/-}BRCA1^{-/-} cells being less sensitive to POLQi than the same cells complemented with BRCA1. Thus, these results suggest that BRCA1-driven resection is an important indicator of sensitization to POLQi. This is another important result with clinical implications for how patients should be stratified for PARPi and/or POLQi treatment.

Major details:

1. Change the title! - very little in text about POLQ addiction and pre-mitotic RPA. Title should convey bulk of the work. Also, title as written seems bit confusing.

We modified the title according to the Reviewer's suggestion.

2. Figure 2 could go into Supplementary(Extended) data.

We appreciate the Reviewers perspective. In our view, the primary purpose of Figure 2 is to demonstrate the distinct cell growth and chromosomal aberrations found in *Polq*^{-/-} with different *Brca1* genotypes. We believe these data are an essential component to the main conclusions, and given we are within the Figure limitations, it should remain a primary figure.

Minor details:

1. Introduction – provide one sentence explaining how BRCA1 counteracts 53BP1-shieldin etc...

We have added a sentence to explain this. See page 3 sentence 5.

2. P4 – “genotypes were readily ‘obtained’ ?” what is the source of these SV40 immortalized cells?

We edited the text to make clear the MEFs were derived from mice in our laboratory (see page 4, paragraph 3, line 6-7).

3. P5 top – “likely producing only a small non-functional peptide”. Did the authors check for POLQ protein by western/Mass Spec?

To the best of our knowledge and after consulting with experts in the Polθ field, there are no reliable Polθ antibodies that currently exist for detecting mouse Polθ. Therefore, Mass spec and western blotting are not viable options. We deleted this statement from the text to avoid speculating.

4. Fig 1A - The cartoon/model schematic is confusing, it looks like BRCA2delExon11 inhibits 53BP1 & Shieldin etc...but the data & text suggests this allele is defective for resection meaning it can't overcome blockade by 53BP1. Same for CC allele, maybe delete the red inhibition line and depict that the two alleles are defective in some other manner.

We modified Fig 1a to clarify the defects of the Brca1 proteins generated by each allele.

5. Fig 1D, either in text or discussion, would be good to discuss that del11 cells can't resect so they don't rely on POLQ (TMEJ is low, supported by data in Fig 4B) perhaps explaining why polq^{-/-} 50% is similar to 50% for WT. While CC cells resect (but can't bind PALB2) and rely on POLQ for viability (TMEJ is higher).

We have edited Fig 1a to indicate which proteins are capable of resection and this point is discussed in detail in paragraph 2 of the discussion, page 12.

6. Are WT cells that lose POLQ sensitive to PARPi? Precedence for this in literature? Ceccaldi et al Nature 2015 show MDA-MB-436 Pol Theta kd same PARPI sensitivity as shScr. How about the cells used in this study?

We assess PARPi response in WT MEFs with and without POLQ in Figure S2c. In line with Ceccaldi, we did not see PARPi sensitivity in BRCA WT POLQ KO cells.

7. P5 bottom – “Taken together, the type of Brca1 mutation strongly influenced Polq-synthetic lethality, whereas when combined with 53bp1 deficiency, both Brca1 mutations required Polq for cellular and embryonic viability.” This sentence is bit confusing. The paragraph is about results in mouse crosses where it doesn’t matter which brca1 allele is used (CC or del11) both are lethal with polq-/- . Is this sentence referring to Fig 1D, where in 53BP1+/+ cell line, del11 colonies are more than CC? Please clarify in text.

We have edited the text to clarify this point (page 5, paragraph 2, line 10).

8. P6 top – “Fig. S2b” should be S2c.

We made this correction in the text (pg 6, paragraph 2 and page 7, paragraph 1).

9. P7 “..suggesting HR is restored to varying degrees, and examined in greater detail in the next section” Confusing, please clarify and where is this examined in next section?

We deleted this comment as we agree it was confusing.

10. P7 – “showing exquisite sensitivity” – 2.3 vs 2.6 fold doesn’t seem like exquisite is right word here.

We have modified the text and removed “exquisite” (page 7, paragraph 3, line 6-7).

11. P7 – “reproducing” should be reproduced

We made this correction in the text (page 7, paragraph 3, line 8).

12. P8 top – “increased sensitivity to ART558”. Maybe add to this sentence, “ and yet, increased resistance to PARPi.

We edited the text and added this wording (pg 8, paragraph 1).

13. P9 – explain one sentence how H2B-mCherry measures mitotic abnormalities.

We have added a sentence to explain this. see page 9, last paragraph.

14. P11- Explain what the BRCA1 mutation is in MDA-MB-436- completely null at protein level?

We have added this information. See page 11, last paragraph.

15. Fig. 2A – are these MEFs from Fig. 1D? Please label cell types in figures.

The MEFs in 2A are derived from 1D as the reviewer anticipated. We have labeled the cell types in the Figure and clarified this in the text (page 6, paragraph 1, line 4).

16. Fig. 3 – please put IC50 label in figure next to Rucarib and ART558. Might help to notate fold change in resistance/sensitivity next to #'s

We added the IC50 label and fold changes to Fig 3.

17. Fig 4A – why didn't HR increase in WT when 53BP1 knocked out? (went down??)

We do not have a concrete answer, but speculate that hyper-resection induced by 53bp1 loss may shift repair preference from HR to TMEJ under the assay conditions.

18. Fig. 4 F – draw an arrow indicating del11 protein. Fig. 4F-G Indicate genotype of cells here – del11/53BP1-/-

We modified Fig. 4 to indicate the del11 protein and genotypes as suggested.

19. Fig.5 – Are these cells 53BP1-/-? Fig. 5C Label the images with MDC1 and RPA. Is Pre-M same as P below? And Post-M is D? If so, label them the same. Fig. 5D – Label which cells are used here.

The cells in Fig. 5 are 53BP1 wild-type and genotypes are indicated in the Figure and legend. We have made adjustments to the labels and added additional labels for clarity and consistency.

20. Fig. 6F – scheme/model is not clear

We modified Fig 6f for clarity.

21. Fig S1D – shouldn't 53BP1 be -/- not +/-?

Correct, thank you for catching that, we made the change.

Concepts to think about:

Fig. 3 – Is it surprising that del11 allele didn't become more sensitive to POLQi (2.3-fold) after 53BP1 deletion? Resection would be upregulated but TMEJ inhibited.

We show that increasing resection by 53bp1 KO resulted in increased HR and TMEJ in the del11 cells (Fig 4A&B), as opposed to inhibiting TMEJ. It is well-established that increasing resection increases TMEJ dependence (Ramsden group Mol Cell 2015, Lord group Nat Comm 2021). We also show that the reason they are not even more sensitive to POLQi is because HR is active in these cells, and depleting HR does result in even greater sensitivity to POLQi (Fig. 4F). We also discuss these points, page 15 paragraph 1.

Whereas CC is already resection competent so might be expected that 53BP1 loss would not change POLQi sensitivity that much?

Although CC are proficient for resection they do not hyper-resect. Whereas when 53bp1 is KO they do hyper-resect (Fig.4C). The Ramsden group, Mol Cell 2015, established that hyper resection induces Polq addiction. Therefore, we believe our data is in line with published findings.

And is it surprising that del11 is sensitive to POLQi at all given that it is defective for resection?

Although they are defective for resection, they likely are able to perform some residual resection. Indeed, to engage Pol theta, the Ramsden group shows only requires 20 bp overhangs, Garcia and Ramsden, PNAS 2020. Thus, we are not surprised that they show some sensitivity, given they are deficient for HR. We add this point to the discussion, page 13 paragraph 1.

Might help to discuss difference between genetic POLQ knockout and pharmacological inhibition of POLQ in text.

We add this important point, see page 14, first sentence.

Reviewer #3 (Remarks to the Author):

In Nature Communications submission NCOMMS-23-35790-T, Kraiss, Johnson and co-authors perform genetic experiments with Brca1 separation-of-function mutant mice and derived cell-lines, to determine the genetic and mechanistic determinants of Pol-theta (POLQ) loss/inhibition-dependent synthetic sick relationship with BRCA1-mutation associated HR-deficiency. Contrasting the genetic synergy between POLQ-loss/inhibition in resection-deficient Brca1 Δ 11/ Δ 11 and Rad51 recombinase recruitment/loading -deficient Brca1CC/CC, across a series of cellular readouts of cell survival, growth, DNA repair and chromosomal stability, they

make a compelling case for DNA end resection proficiency being the critical determinant of PolQ α sensitivity in HR-deficient cells. A major strength of this work is the genetics and data quality, which resolves each phenotype into clear-cut, unambiguous results with the dynamic range to clearly contrast POLQ interactions within distinct yet related backgrounds. Many of the major conclusions are supported by state-of-the-art methodologies, e.g. END-seq to measure resection (complementing RPA IRIF data), and ddPCR method to give absolute measures of pathway usage between the different backgrounds. The PALB2 KO experiment in Figure 6 is also a neat “killer experiment” that successfully attests their major conclusion regarding BRCA1-dependent resection control.

While the work builds on previous findings regarding POLQ α /loss vs HRD synthetic lethality, and the role of resection control (ie via 53BP1 vs resection enzymes), this is a first-time from a BRCA1-mutation centric perspective. This is one of the most clinically-relevant indications for potential POLQ α therapies, and as such the conclusion can be deemed of high scientific interest and medical importance. In my opinion this is a compelling, fascinating and highly convincing study that requires minimal alterations. It should be considered a high-impact/priority manuscript for publication in Nature Communications.

Comments:

1. Synthetic lethality is used quite a lot in the manuscript, and in some occasions, a more nuanced statement (synthetic sick??) might more appropriately describe the genetic interaction.

We appreciate this point raised by the reviewer and added commentary to the discussion to address the idea that synthetic sickness might more appropriately describe the observations (page 12, last sentence).

2. Consider revising Fig1A as it's confusing – from the depiction on the right (a variation on a similar schematic from Nacson et al., Mol Cell 2021), it looks like the Brca1 del11 protein is actually able to inhibit 53BP1 (when they speculate it's the opposite); likewise BRCA1 deltaCC looks like it's an active inhibitor of PALB2-dependent RAD51 loading (which is the wrong way round).

We modified Fig1a for clarity.

3. The statement “Translocation frequency did not increase with ART558, rather, a decrease was observed, although not reaching significance (Fig. 5d, S3c). As expected, ART558 resulted in fewer junctions containing microhomology (Fig. 5e, S3d)” is misleading. If the findings are not statistically significant, the noted effect might simply be down to chance.

We agree with the Reviewer and modified the text (page 11, paragraph 1, line 6-7).

4. Figures S2C and S2D have been incorrectly cited.

We corrected these errors in the text (page 6, paragraph 2 and page 7, paragraph 1).

5. Scale bars in Figure 2B could be included to facilitate accurate comparisons of the chromosomes.

We have added scale bars.